

# Assessing supraglacial lake depth using ICESat-2, Sentinel-2, TanDEM-X, and in situ sonar measurements over Northeast Greenland

Katrina Lutz[1], Lily Bever[1], Christian Sommer[1], Angelika Humbert[2,3], Mirko Scheinert[4], Matthias Braun[1]

[1]Institute of Geography, Friedrich-Alexander-Universität Erlangen-Nürnberg, Erlangen, 91058, Germany
[2]Alfred Wegener Institute Helmholtz Centre for Polar and Marine Research, Bremerhaven, 27570, Germany
[3]Department of Geosciences, University of Bremen, Bremen, 28359, Germany
[4]Institute of Planetary Geodesy, Technical University of Dresden, Dresden, 01069, Germany

*Correspondence to*: Katrina Lutz (katrina.lutz@fau.de)

**Abstract.** Supraglacial lake development in Greenland consists of intricate hydrological processes, contributing not only to surface mass loss, but also to a lowering of the surface albedo and changes in ice dynamics. While the estimation of lake area has recently improved, the determination of the lake volume is essential to properly estimate the amount of water contained in and lost from supraglacial lakes throughout the melt seasons. In this study, four supraglacial lake depth estimation methods, including two new regression approaches, are presented and compared to each other. The first empirical equation is based on

depth information gathered from ICESat-2 crossings over 19 lakes in Northeast and Southwest Greenland, whereas the second empirical equation uses in situ sonar tracks, providing depth information from four lakes on Zachariæ Isstrøm in Northeast Greenland. The depths from both equations are independently correlated to their corresponding Sentinel-2 reflectance values to create empirical relations. The third method is a standardly used radiative transfer model also based on Sentinel-2 data. Finally, the depths for five lakes in Northeast Greenland were derived from TanDEM-X digital elevation models after lake

drainage. All four methods were applied to the five lakes for which digital elevation models were able to be procured, allowing for a direct comparison of the methods. In general, the sonar-based equation aligned best with the estimates from the digital elevation model until its saturation point of 8.6 m. Through the evaluation of the ICESat-2-based equation, a strong influence of lake bed sediment could be seen. The appropriately adapted equation produced slightly deeper depths than the sonar-based equation. The radiative transfer model more strongly overestimated nearly all depths below its saturation point of 16.3 m,

when compared to the digital elevation model results. This large overestimation can be primarily attributed to the sensitivity of this method's parameters. Furthermore, all methods, with the exception of the digital elevation model, were applied to an area in Northeast Greenland on the peak melt dates for the years 2016 to 2022. Finally, a closer look into the uncertainties for each method provides insight into associated errors and pitfalls when considering which method to use for supraglacial lake depth estimation. Overall, the use of empirically derived equations are shown to be capable of simplifying supraglacial lake

depth calculations, while retaining sufficient accuracy under certain conditions.



# 1 Introduction

Supraglacial lakes (SGLs) play an important role in glacial surface mass balance calculations as they collect meltwater and act as conduits for surface and subglacial runoff. The dynamic nature of these lakes is influenced primarily by rainfall, surface temperatures and snowpack thickness (Turton et al., 2021), leading to strong interannual variations in the size and developmental rate of the lakes over the melt season. SGLs, however, are found in surface sinks, which remain in the same locations, due to the influence of bedrock topography on the glacier surface (Gudmundsson, 2003; Lampkin and Vanderberg, 2011), allowing for lake development to be easily tracked. The ability to accurately delineate SGLs in satellite imagery has improved significantly in recent years (Arthur et al., 2020; Dirscherl et al., 2020; Schröder et al., 2020; Dell et al., 2021; Hochreuther et al., 2021; Corr et al., 2022; Lutz et al., 2023). However, while this information provides insight on seasonal lake area trends, the volumes of the lakes are necessary in order to estimate the amount of water that is stored on the glacier and discharged into the subglacial system, in addition to understanding its subsequent impact on the subglacial hydrological system and ice dynamics.

Previously, various methods to measure SGL volumes based on a radiative transfer model have been presented. This method uses the reflectance value of a pixel in combination with estimates of lake bed albedo, optically deep water reflectance, and a two-way attenuation coefficient to determine the depth of the pixel. Originally derived by Philpot (1989), this method has been implemented on SGLs by many research groups on various data sources and areas of interest (Sneed and Hamilton, 2007; Sneed and Hamilton, 2011; Tedesco and Steiner, 2011; Williamson et al., 2017; Williamson et al., 2018; Moussavi et al., 2020; and Arthur et al., 2020). Additionally, research has been conducted to fit empirical functions to in situ data acquired via sonar (Box and Ski, 2007; Fitzpatrick et al., 2013; Legleiter et al., 2014; Pope et al., 2016) or digital elevation model (DEM) (Moussavi et al., 2016) data to achieve better depth estimates. Many of these newly developed algorithms were also compared to the physical radiative transfer model in their analysis; however, the authors' conclusions on the better performing method differ. These varying results could be attributed to the small and, thus, unrepresentative amount of in situ data on which the algorithms were fitted in many of the studies, along with the lack of validation data against which the results can be compared for an objective evaluation. Furthermore, several limitations to these methods are mentioned by various authors. These include the presence of sediment in the water causing depth overestimation (Box and Ski, 2007; Sneed and Hamilton, 2011; Arthur et al., 2020); the effect of wind, and thus waves, on the surface reflectance (Sneed and Hamilton, 2007; Pope et al., 2016; Arthur et al., 2020); and the difficulty of accurately estimating the lake bed albedo and optically deep water (Sneed and Hamilton, 2007; Sneed and Hamilton, 2011; Tedesco and Steiner, 2011; Moussavi et al., 2016; Pope et al., 2016).

With the recent launch of the Ice, Cloud and land Elevation Satellite mission (ICESat-2), a new suite of SGL depth algorithms have begun to be developed, two of which are the Lake Surface-Bed Separation (LSBS) (Fair et al., 2020) and Watta (Datta and Wouters, 2021). Both of these algorithms use ICESat-2's ATL03 laser data product to identify SGL surfaces based on the flatness of the return signal and then automatically estimate the depth along the lake profile. Datta and Wouters (2021) go one step further to create an empirical equation that correlates these lake depths to reflectance values in multispectral satellite



images in order to estimate depths independently of ICESat-2 tracks. These two algorithms were directly compared on a few
test lakes in Fricker et al. (2021), along with several other algorithms created to extract lake profiles from ICESat-2 data, as
well as the physically based radiative transfer algorithm. These results were compared against a manual delineation of the lake
bed from the raw ATL03 data. The physically based algorithm applied to both Sentinel-2 and Landsat-8 red bands consistently
underestimated the manually delineated lake depths, whereas the ICESat-2 algorithms all generally estimated depths within a
certain margin of the manually delineated depths, but contained many dramatic perturbations.

In this study, four supraglacial lake depth estimation methods are compared in order to directly evaluate the behavior and
limitations of each method. These methods include (1) the previously mentioned radiative transfer model (RTM), (2) an
empirical equation derived from ICESat-2 lake crossings, (3) an empirical equation derived from in situ sonar data gathered
in Northeast Greenland, and (4) TerraSAR-X add-on for Digital Elevation Measurement (TanDEM-X) elevation data. The
associated errors and uncertainties of each method are quantified and discussed in order to understand where the pitfalls of the
methods lie. Finally, the methods are applied to the peak lake area extent in the 2016 to 2022 melt seasons in Northeast
Greenland in order to evaluate interannual lake volume trends.

## 2 Data and methods

This study consists of four methods based on various data sources, which will be described in this chapter along with the details
of the derivation of each method. For simplicity, these approaches will be called (1) the radiative transfer model (RTM), (2)
the ICESat-2 equation, (3) the sonar equation, and (4) the DEM method.

### 2.1 Sentinel-2 data

As part of the European Space Agency's Copernicus program, two Sentinel-2 satellites capture multispectral data ranging from
coastal aerosol (442.7 nm) to shortwave infrared (2202.4 nm) with a near daily revisit time in northern Greenland. This high
acquisition rate is advantageous for the monitoring of such a dynamic hydrological process, especially considering the high
frequency of cloud coverage over Greenland's coastal regions, rendering a significant portion of images unusable.
Furthermore, Sentinel-2's red, green, and blue (RGB) bands are provided with a resolution of 10 m, which is valuable for a
detailed analysis of the lakes. The Sentinel-2 images are provided as a top-of-atmosphere product (L1C) or an atmospherically
corrected bottom-of-atmosphere product (L2A) (Drusch et al., 2012). Since the presence of atmosphere in an image would
distort the reflectance value of lake pixels, and thus the depth estimations, L2A images are used in this study. Furthermore, the
cloud-masking algorithm developed by Nambiar et al. (2022) that was specifically created for polar regions is used here to
eliminate cloudy days from the processing chain. All methods except for the DEM method rely on Sentinel-2 data solely or in
combination with other data sources for the estimation of lake depth. Suitable images are ideally acquired from the same date
of any other data acquisitions; however, due to poor atmospheric conditions or missing data, images from the same day may
not be available. In these cases, images from the nearest available date are used.





## 2.2 Radiative transfer model

Developed by Philpot (1989), the radiative transfer model uses a physically based understanding of how light attenuates through the water to provide an estimate for its depth. It is described by Eq. (1):

$$z = \frac{\ln(A_d - R_\infty) - \ln(R_w - R_\infty)}{-g},$$ (1)

where $z$ is depth, $A_d$ is the lake bed albedo, $R_\infty$ is the reflectance of optically deep water (e.g. ocean), $R_w$ is the reflectance value of the lake pixel, and $g$ is a two-way attenuation coefficient. Here, along with the other equations presented in this research, depth below the surface is a positive value. Since the lake bed albedo is unable to be measured directly, the assumption that the surrounding water-free ice can be used as an approximate estimate is utilized. Thus, $A_d$ is calculated from averaging the reflectance values within a 30 m (i.e. 3 pixel) radius around each lake. This radius is used in order to compensate for potential imperfections in the lake masks, which could allow for some water pixels to be included in the $A_d$ calculation. Furthermore, although it is intended that $R_\infty$ be calculated for each image, this is a difficult task due to the fact that the ocean is not present in every scene due to various conditions, such as cloud cover or extensive sea ice presence. Thus, here $R_\infty$ is empirically determined from averaging optically deep water (i.e. ocean) from many Sentinel-2 scenes in the region. Additionally, the values of g are estimated using various relationships of light attenuation in water, the values for which are tuned to the specific wavelength observed by different satellite missions. Here, the values determined in Williamson et al. (2018) for Sentinel-2 are used, specifically 0.1413 for the green band.

## 2.3 ICESat-2

### 2.3.1 Data location

In this study, ICESat-2 is used to define one of the four algorithms. Launched in 2018, ICESat-2 carries a set of six green lasers (532 nm) with a 10 kHz pulse repetition rate (Neumann et al., 2019). This high frequency makes it possible to identify lake profiles with the ATL03 product, where photons are reflected off both the lake surface and bed. Since ICESat-2 has a long revisit cycle of 91 days, it itself is not suitable for the close monitoring of SGL evolution since lakes usually develop and sometimes drain within days to a few weeks. Since Sentinel-2, however, has a high temporal and spatial resolution in Greenland, such a monitoring task is enabled. Thus, the depths derived from the ICESat-2 lake crossings are correlated with temporally coinciding Sentinel-2 images to create a depth-reflectance relationship.

Figure 1(a) shows the locations in Greenland for which an ICESat-2 path crossed a filled supraglacial lake, depicted by purple points. In order to allow a sufficient number of ICESat-2 lake crossing and Sentinel-2 imagery, we augmented our data set from Northeast Greenland with datasets in Southwest Greenland. In total, 19 lake crossings over the 2019 to 2022 melt seasons were found in which the lake profiles contained enough points for the lake surface and bed to be distinguishable. These crossings are depicted in Fig. 1(c) for the lakes found in Northeast Greenland and (d) for those found in Central-West and Southwest Greenland. For each lake crossing, a corresponding Sentinel-2 image is acquired from the same day or the closest



day to the ICESat-2 crossing as possible. In Table A1, detailed information for each lake crossing is listed, including the date of acquisition, the ICESat-2 beam ID, the number of lakes acquired from each track and the corresponding Sentinel-2 image used for further processing.

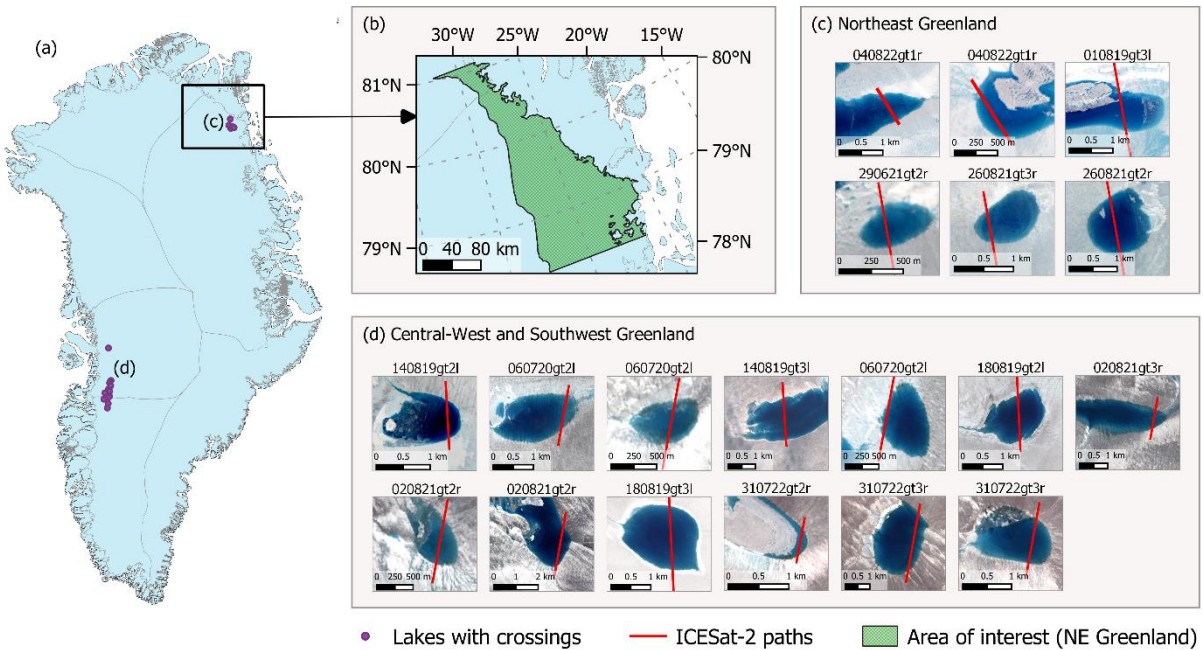

**Figure 1: (a) An overview of Greenland, showing the locations of the supraglacial lakes with ICESat-2 crossings used in this study (purple points) labeled by (c) and (d), which are shown in further detail in the respective subimage. The gray lines represent Greenland's basin boundaries, produced by Rignot et al. (2011). (b) A closer view of Northeast Greenland, with the region over which an interannual analysis will be produced. (c) and (d) show the ICESat-2 track paths (red) for the lakes marked in (a) for Central-West/Southwest Greenland and Northeast Greenland, respectively. Each lake is labeled with its acquisition date and ICESat-2 beam ID and is represented visually with the Sentinel-2 image closest to the acquisition (listed in Table A1).**

### 2.3.2 ICESat-2 lake crossing track retrieval

Here, ATL03 tracks from ICESat-2 are used to gather a set of SGL depth profiles. Due to ICESat-2's long revisit time and narrow footprint, a lake crossing is a relatively rare event. Areas in Northeast and Central-West/Southwest Greenland (shown in Fig. 1) were manually investigated over the 2019 to 2022 summer melt seasons to identify potential lake crossings of SGLs using NASA's OpenAltimetry tool (https://openaltimetry.org/data/icesat2/). The date, geolocation and track ID of unfrozen and high quality lake crossings were then entered into the Jupyter Notebook processing tool developed by Fricker et al. (2021) (https://github.com/fliphilipp/pondpicking). In this tool, the ICESat-2 ATL03 data is shown in an editable window, where the lake surface and bed can then be manually drawn. An example from one of the lakes can be seen in Fig. 2(a), where the manually drawn lake surface is depicted by the blue line and the lake bed by the red line. It should be noted that these lines are not determined in regard to the photon confidence level (Neumann et al., 2019), but rather by the density of photon return signals. Based on best judgment and consistency with previous studies, the surface and bed profiles were drawn along the areas





of highest photon concentration, typically just below the first appearance of photon accumulation. However, the width of the area of high photon concentration spans an average range of 0.62 m for the lake surface and 1.06 m for the lake bed, resulting in a significant difference in where the surface or bed could be delineated. Figure 2(b) shows the corresponding track path over a Sentinel-2 image from the previous day. The Sentinel-2 L2A images are downloaded and then preprocessed by converting the digital numbers to reflectance values. In this study, all three RGB bands are investigated to determine which band produces the most reliable depth results. Thus, reflectance values are collected for each band along the profile of each lake.

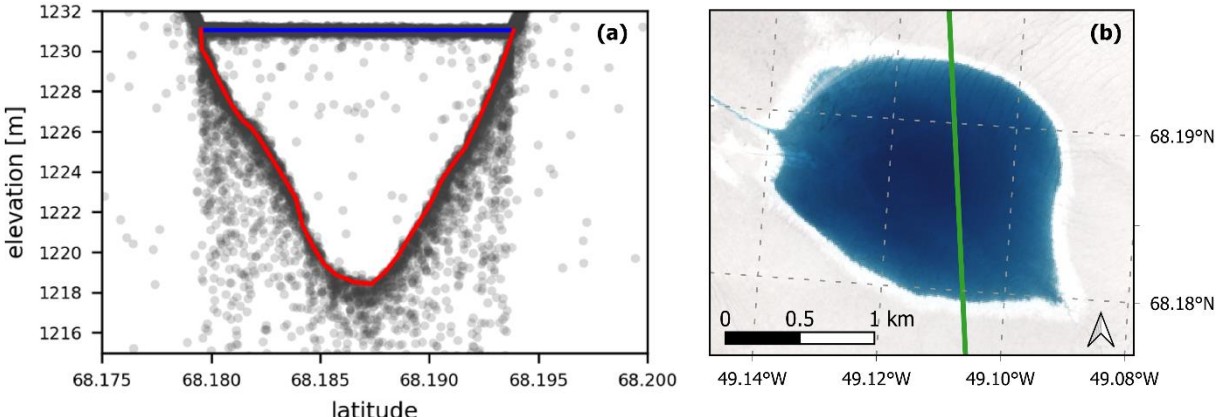

**Figure 2: An example SGL in southwest Greenland captured on 18 August 2019. (a) Plotted ICESat-2 ATL03 data showing the lake surface (blue) and lake bed profile (red), created using the picking tool by Fricker et al. (2021). (b) the Sentinel-2 image from the previous day, showing the ICESat-2 path crossing the lake (green).**

### 2.3.3 Lake depth equation

Firstly, a refraction correction needs to be applied to the ICESat-2 depths to account for the change in speed of light in water. As used in Parrish et al. (2019), it is defined as Eq. (2):

$$R = \frac{Sn_1}{n_2} \ ,$$

(2)

where $R$ is the adjusted depth, $S$ is the uncorrected depth, $n_1$ is the refractive index of air ($n_1 = 1.00029$), and $n_2$ is the refractive index of green light in water ($n_2 = 1.334$). Then, the corrected depths and corresponding Sentinel-2 reflectance values for all 19 lake crossings are plotted together for each of the red, blue and green bands to determine which band exhibits ideal attenuation behavior. An exponential function is then fit to each band individually, the $R^2$ values of which are then evaluated to determine which band is most suitable for accurately estimating depth from reflectance values.

### 2.4 In situ sonar

One of the depth algorithms is based on sonar data gathered in situ in Northeast Greenland. For this, a self-built remote controlled boat equipped with a sonar sensor was constructed. This boat consists of a floatation board, two propellers, a



waterproof box containing electrical wiring and a battery, as well as a Lawrence Elite 7 FS sonar sensor and corresponding
monitor, seen in Fig. 3(a) and (b). During fieldwork in July 2022, four depth profiles were measured with this boat, the locations
of which are depicted in Fig. 3(d). These lakes are located upstream of the grounding line of the glacier Zachariæ Isstrøm, as
shown in Fig. 3(c). These lake profiles are then processed using the software Reefmaster 2.0, where the lake bed is manually
delineated from the sonar signal and then converted into vector data points. Some error could arise from the delineation of the
lake bed from the sonar plot, as there is not only some noise in the backscatter but also limitations in manually extracting the
surface. This error is estimated to be 0.20 cm. As a note, the naming convention of these lakes is based on the location of
topographical sinks in the Northeast Greenland region highlighted in Fig. 1(b). Within some sinks, multiple untouching lakes
regularly form, requiring the use of the descriptors *a*, *b*, *c*, and *d*.

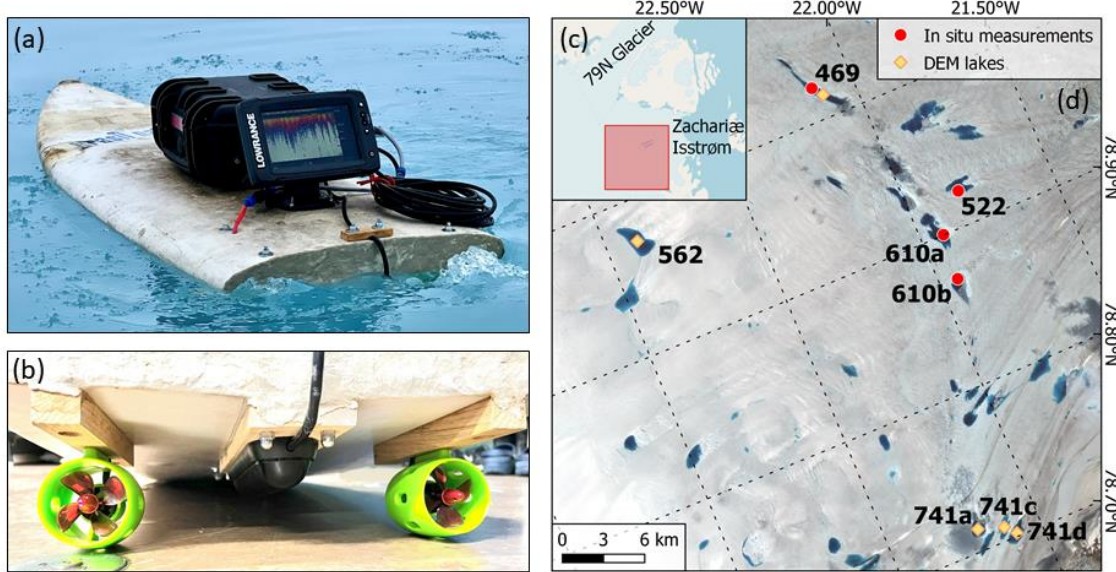

**Figure 3: (a) The remote controlled sonar-equipped boat used in this fieldwork to gather lake depth profiles. (b) The underside of**
**the boat, showing the sonar sensor in the middle along with the two propellers. (c) A zoomed out view of the scene shown in (d),**
**showing the relation of the measurements to the two major glaciers in Northeast Greenland. (d) A Sentinel-2 image from 19 July**
**2021 with the sites on which the in situ measurements of four supraglacial lakes were taken (marked with red points), and the lakes**
**for which DEMs were created (marked with yellow diamonds). Each lake is labeled with its ID number.**

The four sonar tracks acquired in situ via a remote controlled boat are displayed in Fig. 4, where the tracks are overlaid onto a
Sentinel-2 image captured from the date before the sonar acquisition. Each track contains data up to around 100 m offshore.
The tracks in Fig. 4(b)-(d) show depths up to roughly 7 m deep, while the track in Fig. 4(a) shows depths above 10 m with a
small area around 14 m deep.



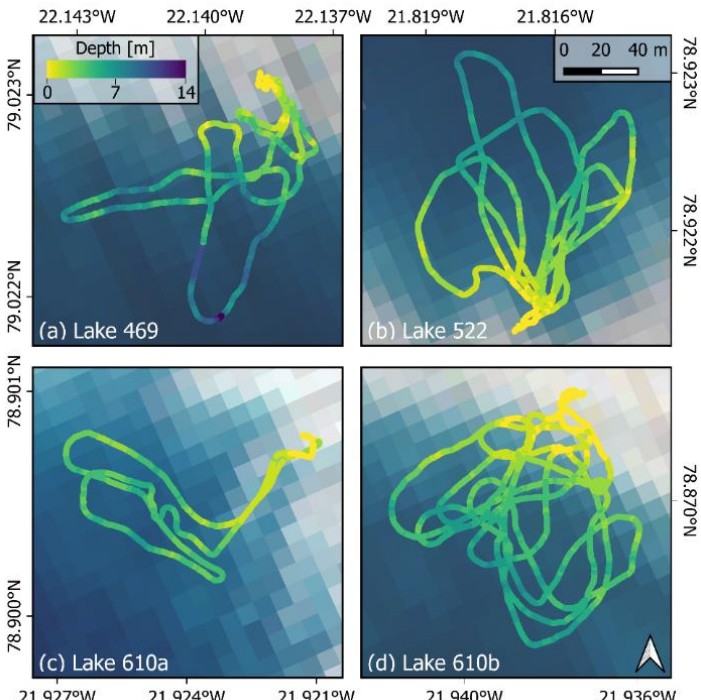

**Figure 4: The sonar tracks captured over the four SGLs acquired in situ with a remote controlled boat. The backgrounds are Sentinel-2 L2A RGB images from one day before the sonar acquisition date.**

The depths from these sonar tracks are correlated to the Sentinel-2 imagery to create a depth–reflectance relationship. The sonar acquisition dates and Sentinel-2 tile IDs can be found in Table A2. The sonar depth data are acquired at a much higher resolution than the resolution of Sentinel-2 images, so an average is taken of the sonar depth data over every Sentinel-2 pixel. This results in an average standard deviation of 0.32 m among all the pixels used to create the equation. The sonar tracks, however, did not always pass perfectly through the center of the pixel, so the measured depths may only be representative of a portion of the pixel. An exponential function is then fit to the depth–reflectance data of the most suitable band, determined in Section 2.3.3.

## 2.5 TanDEM-X

In 2010, the TanDEM-X mission was launched, creating a configurable high resolution spaceborne radar interferometer in the X-band. Synthetic Aperture Radar (SAR) DEMs of Northeast Greenland are created from Co-registered Single look Slant range Complex (CoSSC) data based on differential interferometry (Sommer et al., 2022). Initially, interferograms are calculated from concatenated SAR acquisitions in the along-track direction. Thereafter, the differential phase of each interferogram is unwrapped using a minimum cost flow algorithm and converted to elevation values above a reference surface. As the reference DEM, we use the global Copernicus DEM GLO-30 with a spatial resolution of 30 m (European Space Agency





2022). Eventually, each newly created TanDEM-X DEM is iteratively co-registered to the Copernicus DEM in the horizontal and vertical plane to remove remaining systematic offsets or geometric distortions. The co-registered DEMs, captured after the complete drainage of a supraglacial lake, are used to determine the bathymetry of the lake. From this, the lake depths can be determined for a previous date when the lake was filled.

Since TanDEM-X coverage is sporadic due to the campaign-based DEM acquisition (Bachmann et al., 2021) and complete lake drainages are relatively infrequent, the acquisition of a post-drainage lake DEM is a relatively rare event. Nonetheless, post-drainage DEMs were able to be created for five lakes in Northeast Greenland over the 2021 melt season, the locations for which are shown in Fig. 3(d). In order to determine the lake depth, a lake surface elevation is determined from the boundary of the segmented lake mask from a date before drainage (Lutz et al., 2023). The elevations around the boundary are averaged

to produce one surface value, from which the elevations of the lake bed are subtracted, resulting in the lake depth at each pixel. Since the resolution of the DEM is 10 m, some variation in the average elevation found within lake edge pixels is to be expected, especially around strongly sloped or rugged areas. When determining the inclusion of a lake in this study, any lake with a surface elevation RMSE of more than 1.5 m was excluded. Nonetheless, of the five lakes evaluated in this research, the average RMSE for the surface elevation was 0.94 m.

**2.6 Method comparison**

In order to evaluate these lake depth estimation techniques, all four methods are applied to the five lakes for which DEMs were procured. Sentinel-2 imagery is chosen as close to the drainage date of each lake as possible. The data pertaining to the lake drainage dates, the DEM acquisition dates, and the Sentinel-2 imagery used are detailed in Table A3.

Additionally, the sonar equation, the ICESat-2 equation and the RTM method are applied to peak melt dates in the 2016 to
2022 melt seasons over an area in Northeast Greenland encompassing the Nioghalvfjerdsbræ (also known as the 79°N Glacier) and Zachariæ Isstrøm glaciers. This region can be seen in Fig. 1(b). The dates for maximum lake area extent are determined from the results found in Lutz et al. (2023). These three methods are then applied to the lake area extent derived from their method. This allows for a comparison of the methods on a large scale while also showing the interannual variability of the meltwater development in the region.

**3 Results**

**3.1 ICESat-2 depth equation**

Figure 5(a)-(c) displays three plots, one for each RGB band, in which the depth values gathered from 19 ICESat-2 lake profiles are plotted against their corresponding Sentinel-2 reflectance values. For each band, the data shows two distinct trends, correlating to the region from which the ICESat-2 data was acquired, i.e. whether the lakes were located in Northeast or
Southwest Greenland. Due to such distinct behavior between the regions, two curves were fit to the data for each band, where the orange data points and curves reference data from Northeast Greenland and the green ones reference data from Southwest



Greenland. The ice in Southwest Greenland is generally more heavily covered by sediment than Northeast Greenland, which can be seen by the darker color of the ice surrounding lakes in Figure 1(d). This presumably explains the shift in depth measurements towards lower reflectance values in data from Southwest Greenland. The distinction between the regional curves

becomes stronger with larger wavelengths, i.e. the curves are the most distinctly separated for the blue band and the least separated for the red band, implying a stronger influence of the sediment for wavelengths that penetrate deeper into the water. Furthermore, the red band, shown in Fig. 5(c), shows clear limitations due to attenuation. Here, reflectance values are only able to estimate depths up to around 3 m, depths above which are represented by similar reflectance values. Due to this behavior, curves for the red band were only fitted on data up to 3.5 m. In contrast, the data points for the green band only start

stacking once they reach around 10 m deep, while the rest are distributed fairly evenly across the higher reflectance values.

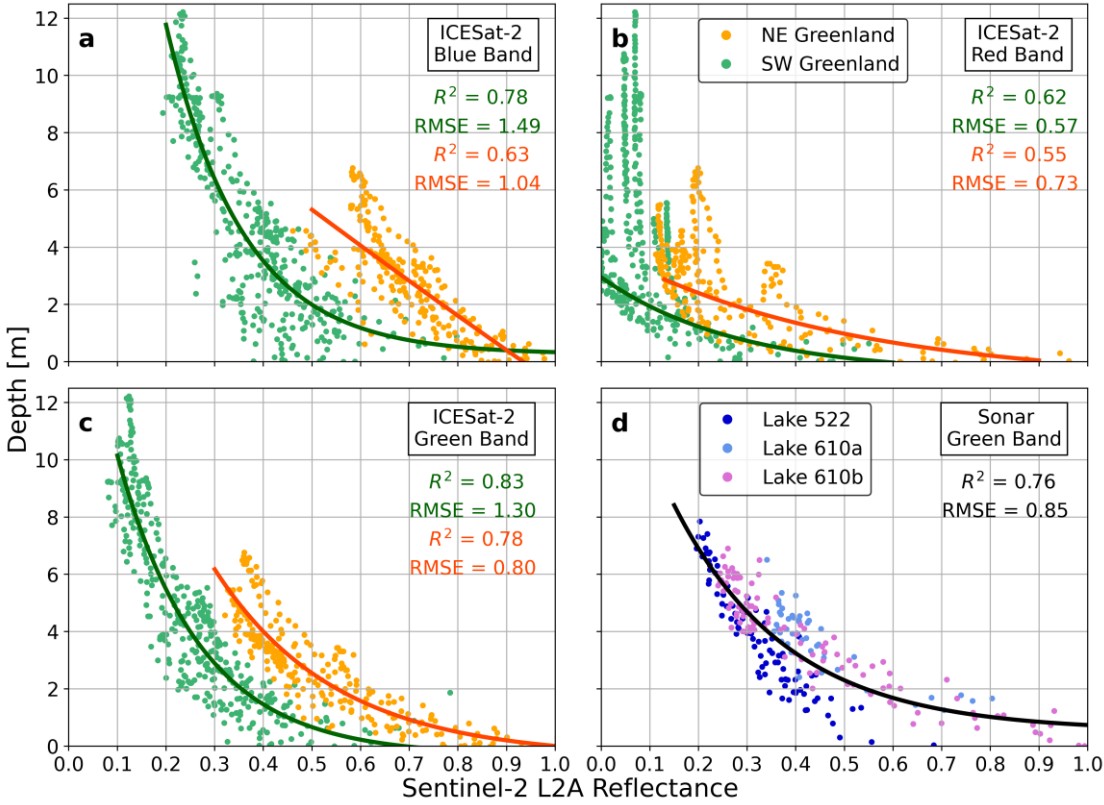

**Figure 5: Depth versus reflectance plots for ICESat-2 depths (a)-(c) and sonar depths (d) against their corresponding Sentinel-2 L2A reflectance values. For (a)-(c), data gathered from ICESat-2 tracks in Southwest Greenland are represented by green points and data from Northeast Greenland are represented by orange points. For (d), each color represents a different lake from which the**

**sonar data was gathered. For (a)-(d), the respective curves represent exponential equations fit to the data from the specified band and region, the RMSE and R² values for which are listed in each subplot.**





Since the green band has the best R² values and has a fairly consistent depth–reflectance ratio across nearly the full spectrum, it has been selected as the basis for this depth algorithm. The best fit to the data was found using an exponential function, defined by Eq. (3) and Eq. (4) for the Southwest and Northeast regions, respectively:

$$z_{SW} = 18.8999e^{-5.9037x} + 0.3237 \ , \tag{3}$$

$$z_{NE} = 21.9222e^{-4.0180x} + 0.3902 \ , \tag{4}$$

where $z$ is the lake depth and $x$ is the Sentinel-2 L2A reflectance value. These functions are plotted over the gathered ICESat-2 data in Fig. 5(c). The root mean squared error (RMSE) and the coefficient of determination ($R^2$) values for the Southwest function are 1.30 m and 0.83, respectively, while for the Northeast function they are 0.80 m and 0.78, respectively. To better 260 assess the uncertainty variation along the curve, the RMSE was calculated for bins of 0.05 increments over the reflectance values, since how well the curve fits to the data varies across the reflectance values. Here, the RMSE values ranged from 0.54 m to 1.75 m for the Northeast function. While the data points in the Southwest function include depths up to 12 m, depths only up to around 7 m were gathered in the Northeast. This allows the Southwest equation to be reasonably valid up to roughly 10 m of depth, where the number of samples declines and the depth values start to saturate at similar reflectance values. The 265 Northeast equation, however, can only be reasonably used to depths up to around 6 m.

## 3.2 Sonar depth algorithm

While post-processing the sonar data, depths were compared at points where the boat passed more than once, where the difference should in theory be zero. While Lakes 522, 610a, and 610b had an average crossover difference of 0.11 m, the average difference of the crossover points for Lake 469, was 0.68 m, with differences found up to 2.11 m. The large 270 discrepancies can be attributed to the rough water conditions rocking the boat during data acquisition. This lake was, thus, removed from the analysis. The relatively small discrepancies found for the other lakes could be attributed not only to minor fluctuations in the lake's surface, but also to the precision of the sonar sensor and the geospatial sensor. The depths from these three sonar tracks were plotted together against their corresponding Sentinel-2 reflectance values for the green band, as seen in Fig. 5(d). Since all three lakes were located in Northeast Greenland, the data follow one trend. An exponential equation was 275 fit to the data, which is described by the Eq. (5):

$$z = 14.9572e^{-4.2629x} + 0.5242 \ , \tag{5}$$

where $z$ is the lake depth and $x$ is the Sentinel-2 L2A reflectance value. The data points corresponding to the Sentinel-2 green band are plotted in Fig. 5(d), where the tracks along each lake are a different color. Here, the RMSE for the fit equation is 0.85 m and the $R^2$ is 0.76. Furthermore, the quantification of the uncertainty was handled similarly to the ICESat-2 equation by 280 calculating the RMSE for 0.05 increments over the reflectance values. Among these bins, the RMSE ranges from 0.27 m to 0.94 m.



### 3.3 Comparison of SGL depth estimation methods

The five lakes for which DEMs were procured are shown in Fig. 6. Here, all four depth estimation methods are shown. Some areas of the DEMs are marked as invalid since Sentinel-2 imagery showed some water remaining on the lake bed after the
drainage. These areas, while shown in the other methods, were not used in the calculation of volumes, maximum depths or errors to allow for a consistent comparison. Using the DEM results as reference, the limitations of the other three methods can be seen. While the sonar equation tends to produce the shallowest results, they are the results most in agreement with the DEM estimates, up to its saturation depth of around 8.6 m. The sonar equation produces the largest errors in shallower regions, which is exemplarily shown for Lake 741c, where the maximum depth, estimated by the DEM method, is 5.83 m. Here, the sonar
equation overestimates the lake's volume by 62.5%. While this method overestimates shallow areas (< 3 m), it produces results similar to the DEM for depths between 3 and 7 m deep. The ICESat-2 method, however, overestimates depths across the entire depth range, until its saturation point of around 12.7m. The lowest volume errors in comparison with the DEM estimates are found with Lake 469, where the ICESat-2 method only overestimates the total volume by 6.0%. While this error is low, it is unrepresentative of the comparison of individual depths. The majority of the lake is overestimated, whereas the deeper areas
(13 to 27 m deep) are underestimated. Since the data used to fit both the sonar and ICESat-2 methods is limited to shallower depths, the behavior of both methods over 6 m is unconstrained by actual data and thus most likely deviates from optimal estimates.

The RTM method even more strongly overestimates the lake volume for all five lakes, when compared to the DEM results. For shallower lakes, e.g. 741c and 741d, the RTM method overestimates the volume by 137.5% and 75.4%, respectively. This
method has its lowest error for the largest lake (469), with an overestimation error of only 6.7%. Similar to the ICESat-2 estimates, however, this is unrepresentative of the accuracy of individual depth estimation. The RTM method overestimates depths with increasing error as the depth increases, until it reaches a saturation level around 16.3 m. For Lake 469, the majority of the lake is significantly overestimated with the deeper areas (18 to 27 m deep) underestimated. A closer look at the comparison of DEM depth to the depths derived from the other methods can be seen in Fig. B1.







**Figure 6: The four lake depth estimation methods (DEM, sonar equation, ICESat-2 equation, and RTM) are applied to five different lakes (469, 562, 741a, 741c, and 741d). The total volume estimated from each method is shown for each lake, along with the maximum estimated depth. Areas where the DEM could not be calculated due to residual water are marked by pink. These areas, while shown, were excluded from the calculation of the volume and maximum depths for the other methods.**





## 3.4 Interannual comparison of peak melt extent

Figure 7 shows the volume estimates for the ICESat-2 equation, sonar equation, and RTM method for the dates of maximum spatial extent over the 2016 to 2022 melt seasons over the area in Northeast Greenland shown in Fig. 1(b). The uncertainties associated with each method are shown via error bars. The 2018 melt season in Northeast Greenland has been shown to be comparably dry and cold (Turton et al., 2021), which is reasonably reflected by the significantly lower volume estimates from all three methods. This large difference can be seen for example when comparing the largest volume for the sonar method in 2016 with 0.903 km$^3$ of total water, whereas the estimates for 2018 were less than half of that with only 0.349 km$^3$. Besides this large deviation, the interannual variability of the total amount of meltwater gathered in SGLs is rather low, considering the large span of the error bars. The RTM estimates have the largest interannual variability with a standard deviation of 0.287 km$^3$, compared to 0.194 km$^3$ and 0.167 km$^3$ for the sonar and ICESat-2 equations, respectively. This can be explained by the larger saturation depth inherent to the RTM method. With a larger range of potential depths, the amount by which estimates can vary increases.

When comparing the variability of lake area in Lutz et al. (2023), the interannual lake area variability is much larger than the interannual lake volume variability. For example, the 2016 lake area extent was 346.5% larger than the 2018 extent, whereas the 2016 total lake volume is only 158.7% larger than the 2018 total lake volume, based on estimations from the sonar equation. This suggests that more lake area is rather easily gained but is composed of relatively shallow water, resulting in less volume change. A closer look at the distribution of average lake depths per year can be seen in Fig. B2.

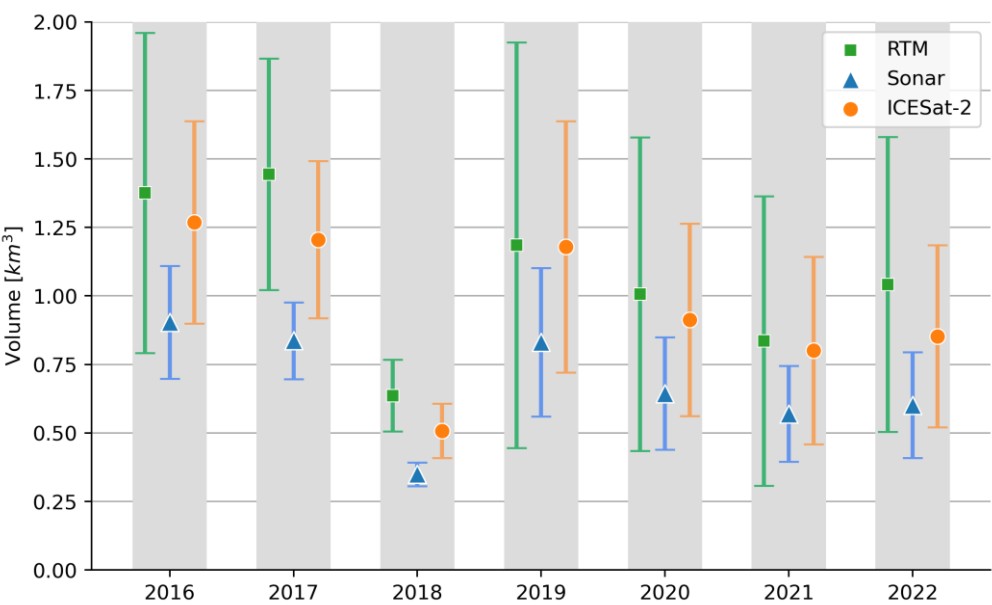

**Figure 7: The total lake volume [km$^3$] over Northeast Greenland (area defined in Fig. 1(b)) for the date of peak melt area determined in Lutz et al. (2023) over the 2016 to 2022 melt seasons. The estimates from the sonar equation, ICESat-2 equation, and the RTM method are shown for each melt season, along with their estimated uncertainties.**



## 4 Discussion

### 4.1 Usefulness of the different visible bands for depth analysis

Throughout literature, both the red and green bands have been used for single-channel depth estimation in multispectral imagery. In our study, the saturation of the red band with depths of around 4 m is clearly seen (see Fig. 5(b)). This has also

been noted by several other research groups (Datta and Wouters, 2021; Moussavi et al., 2016; Pope et al., 2016; Williamson et al., 2018); however, the red band is often used despite this. Some studies used the red band without justification (Box and Ski, 2007; Fitzpatrick et al., 2013), while others concluded that the use of the red band resulted in better performance based on a comparison of the depth estimations from two different satellites (Williamson et al., 2017; Williamson et al., 2018). While there is a clear depth-reflectance trend up to around 3 m deep, this limits the scope of such a method. From our sonar, ICESat-

2 and DEM data sources, it can be seen that lake depths are often over 5 m deep, with all five lakes showing DEM-derived depths between 10 and 25 m. Moreover, in the interannual comparison, between 8.1% (in 2020) and 32.1% (in 2018) of lakes had an average depth larger than 4 m over the melt seasons (see Fig. B2). While the majority of lakes were quite shallow, a significant portion of the water volume is present in the lakes where average depths are larger. Based on this, the use of the green band seems to be a more suitable choice for estimating deeper lake depths, which was similarly determined by Sneed

and Hamilton (2007), Sneed and Hamilton (2011), and Tedesco and Steiner (2011). However, an analysis by Pope et al. (2016) was conducted comparing red and green estimates to depths derived from digital elevation models, which found that the green band overestimated the lake depths when the radiative transfer model was used. As seen through our study though, estimates can vary quite strongly depending on the method used, and the radiative transfer model is particularly prone to overestimating lake depth. Moussavi et al. (2016) used DEMs to define several lake depth equations in comparison with the radiative transfer

model. In their study, the green band performed best for both single-channel equations; however, they concluded the use of the red band was preferential due to the lower sensitivity of the red band to variations in the radiative transfer model parameters. While the red band may be better suited to the shallow depths, the advantages of using the green band in single-channel depth estimation methodologies seem to outweigh the disadvantages.

### 4.2 Differences and potential errors in the methodological approaches

Since each method is derived from different data sources and is dependent on various variables, the uncertainties present in each method can contribute to the discrepancies seen among the depth estimations. Firstly, there are a couple effects inherent to regression equations. Since the data for both the sonar and ICESat-2 equations are limited in the regard that they do not contain deep (> 7 m) nor many very shallow (< 0.5 m) depths, the regressions are not properly bound at the extremes. This effect can be seen in the sonar equation's inability to estimate depths above 8.6 m as well as in the overestimation of depths in

the ICESat-2 equation. Furthermore, the addition of deeper data points for both the sonar and ICESat-2 equations could affect the curvature of the entire regression, which would affect the estimation of the rest of the depths as well. Moreover, the inability of both equations to accurately estimate very shallow areas is apparent in Fig. 6, where the edges of the lake are never estimated



to be as shallow as they are in the DEM estimates. This can be attributed to the sonar equation regression never reaching a value below 0.5 m in a physically meaningful range, rendering this method incapable of estimating depths below this value.

In the ICESat-2 equation, however, the regression reaches zero, but only at a very high reflectance, which is less likely to be seen in shallow lake edge waters. This difference seen in Fig. 6 could also be due to an inaccurate estimation of the DEM's surface level, which would be more apparent in shallower areas, since the average RMSE among all five lakes for the surface elevation is 0.94 m. Additionally, several studies have reported that after a rapid drainage, local ice uplift has been observed (Chudley et al., 2019; Das et al., 2008; Doyle et al., 2013; Hoffman et al., 2011). While the maximum observed uplift was 1.2

m (Das et al., 2008), most groups reported that the ice slowly settled back to a lower elevation up to 0.2 m above the pre-drainage elevation. Thus, DEMs created after a rapid drainage could potentially still contain a vertical offset, which could affect the comparison to other methods.

Furthermore, the geolocation of ICESat-2's photons could introduce inaccuracies due to horizontal accuracy and footprint size, resulting in a mismatch between depth and reflectance information. A geolocation error between 2.5 and 4.4 m was reported

through validation with ArcticDEM (Luthcke et al., 2021), which is below the specified ATLAS photon horizontal geolocation of 6.5 m ($1\sigma$) (Neumann et al., 2019). However, each beam has a nominal footprint diameter of 17 m, which is larger than the spatial resolution of Sentinel-2 (10 m) and might result in an inaccurate comparison of ICESat-2 depths and Sentinel-2 reflectance.

Next, the depth overestimation in the RTM method seen throughout the lakes in Fig. 6 can be attributed to the difficulties

involved in the calculation of the parameters $A_d$, $R_\infty$, and $g$. Firstly, the reliance on an estimation of $R_\infty$ by averaging ocean pixels from other scenes can introduce errors due to a potential difference in atmospheric conditions and sun elevation, among others. The ocean itself also inherently has a relatively wide spread of reflectance values. Secondly, using the lake edge reflectance, $A_d$, as a proxy for lake bed albedo can introduce errors into depth estimation. Tedesco and Steiner (2011) found this approximation to lead to average depth errors of 15.9% when estimating $A_d$ with the green band. Similarly, it was found

by Moussavi et al. (2016) that $A_d$ estimates based on lake edge reflectances were higher than optimized lake bed albedos by 5 - 10%. This then translates to around a 20% depth underestimation in green bands. Additionally, the calculation of $A_d$ can be skewed by imperfect lake masks. If the drawn lake boundary does not actually follow the edge of the lake, the $A_d$ value would be calculated from lake pixels instead of just the surrounding ice. If $A_d$ were calculated from water pixels, this would lead to a shallower depth estimation. This could also be problematic for situations in which an SGL is located nearby a non-ice feature,

such as a nunatak. Finally, the theoretical estimation of the variable $g$ could improperly reflect the situation in reality. Pope et al. (2016) state that $g$ for green bands is more sensitive to errors than the red band, since green light attenuates through water more slowly. This fact, along with the differences in lab-based and theoretically calculated $g$ values, implies a strong influence of variations in this value on the estimated lake depth. Thus, for all three parameters discussed here, a sensitivity study was conducted on Lake 562, more details for which can be found in Fig. B3. It was found that a change of 0.01 m$^{-1}$ in the variable

$g$ resulted in a 7.4% change in estimated volume or a 0.60 m change in average depth of the lake. Furthermore, the variable $R_\infty$ was evaluated over the span of reflectance values found in the nearby ocean. Over this reflectance span of 0.05, there was





a difference of 14.7% in the resulting volume estimations. Additionally, since the values for $A_d$ are calculated on the pixels surrounding each lake, the width of the area around the lake is considered here. In this instance, lowering the distance to 10 m or raising it to 60 m had some effect, but it was smaller than the effect seen with the other two variables. This, however, could potentially vary significantly for other lakes around which the ice surface is more dynamic, e.g. with sediment dispersion. Overall, the sensitivity of these three variables, as well as the rough estimation of some, contribute to the tendency of the RTM to produce erroneous results, without the variables having been tuned to specific scenarios.

### 4.3 Limitations of lake depth estimation from multispectral images

While methodologies employing multispectral images for the purpose of estimating SGL depths have been shown to work well for most situations, there are certain limitations of such methods which must be acknowledged. Firstly, sediment is deposited on the surface of the glacier, which can then enter the supraglacial lakes and settle to the lake bed, as seen in Fig. 8(a), appearing as dark regions. This is further exemplified in Fig. 8(b), which shows the amount of sediment left behind after a lake drainage. When estimating lake depth from reflectance values, these areas would be measured with a very low reflectance, which would then be falsely translated as much deeper than in actuality. Even though the ice in Northeast Greenland is relatively clean, sediment is still prone to gather in some lakes in the region. Due to the insights gained from the ICESat-2 analysis (see Fig. 5), it can be assumed that the effect of sediment in Southwest Greenland is even more pronounced. A second situation inducing errors in depth estimation is frozen lake surfaces. Figure 8(c) shows an image of a lake with a frozen, but not snow covered, surface layer. When the surface is frozen but still transparent, it increases the reflectance value in the satellite image. The difference in color between a frozen and unfrozen surface can be seen in Fig. 8 (d), where a small portion of the surface is unfrozen. From satellite images and even at a high helicopter flying height, it is not noticeable that these lakes are frozen. The consequence of this is that there is little to no indication in satellite images that the surface is frozen and the lake will be estimated as shallower than in actuality. Furthermore, if there are any shadows (e.g. from clouds, surrounding topography or internal topography), this will influence the reflectance value, causing the depth to be overestimated. Contrarily, if there are thin clouds or fog present over a lake, this could make the lake color appear lighter, thus causing depth underestimation. This also highlights the importance of using bottom-of-atmosphere products (e.g. Sentinel-2 L2A) to minimize atmospheric effects. Finally, the presence of floating ice inhibits the estimation of lake depth underneath it, causing these lake volumes to be severely underestimated if there is wide ice coverage. This also can be problematic in time series analysis, as the floating ice tends to shift around the lake. What could be perceived as a large increase in lake volume could in actuality be the floating ice shifting from covering up a deep part of the lake to a shallower part.





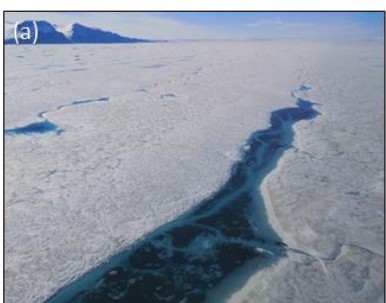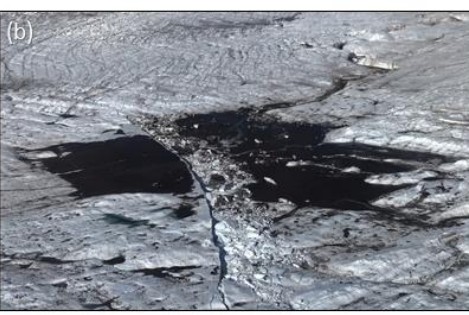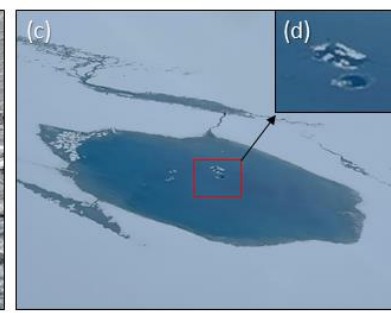


**Figure 8: Aerial images of sediment-filled, drained, and frozen SGLs in Greenland. (a) A sediment-filled SGL in Northeast Greenland, photo taken in July 2022 by M. Braun. (b) The sediment left behind after an SGL drainage in Northeast Greenland, photo taken in July 2022 by A. Humbert. (c) An SGL in Scoresby Sound with a frozen surface with the exception of a small portion highlighted in (d), photo taken in August 2022 by K. Lutz.**

## 5 Conclusions and outlook

Throughout the development and implementation of these four supraglacial lake depth estimation methods, it can be seen that each method has certain areas of suitable applicability. The reliability of DEM differencing is advantageous for understanding the full bathymetry of a lake, which cannot be dependably obtained through the other methods, especially for deeper lakes. As long as an accurate surface elevation can be estimated, this method is useful for a closer evaluation of individual lakes, but is

not suitable for long-term or widespread monitoring due to limited acquisitions of TanDEM-X and the irregularity of complete lake drainages.

    The other three methods presented here, however, would be more suitable for lake volume estimation on a larger scale. The radiative transfer model is standardly used due to its reliance purely on optical data and incorporation of the properties of surrounding features. This allows it to be more useful for widespread monitoring; however, the sensitivity of its parameters

can easily cause an overestimation of depths. Due to the difficulty in properly estimating these parameters, the use of a more simplistic equation could be preferential. Even though the data directly obtained from ICESat-2 or in situ sonar devices are impractical for the continuous monitoring of lakes, the correlation of their depth data to optical missions with a high revisit rate, such as Sentinel-2, allow for a simple and direct estimate of lake depth in optical imagery. The sonar-based equation, while limited in use to depths below 8 m, seems to fit the DEM estimates best. Through the evaluation of the ICESat-2 depths

on different Sentinel-2 bands, the influence of the lake location (Northeast vs. Southwest Greenland), is quite apparent. This distinction in the data is most presumably due to the higher percentage of sediment on the ice in many parts of Southwest Greenland, causing a shift to lower reflectance values. Through the band analysis, the green band appears to be most suitable for general applications due to its good depth-reflectance ratio and higher saturation limit. However, to improve the methodology overall, combining estimations from red, green, and blue bands into a single algorithm could potentially

overcome the attenuation limitations of each band, allowing for more accurate estimations in shallow water with the red band and deeper water with the blue. The limitations of a method based purely on multispectral images, however, will still be present.



In order to improve both the ICESat-2 and sonar equations, the acquisition of more depth data would be required. Not only would more data reduce the uncertainty attributed to the regression fit, but the acquisition of data with deeper depths would allow the equations to be properly extended to depths above their current limitations. Additionally, the acquisition of in situ data during a simultaneous ICESat-2 passing would allow for a direct comparison of the raw data on which both methods are respectively based. To acquire data from larger portions of a lake than is feasible with a remote controlled boat, the use of airborne lidar could be advantageous. Overall, this study shows the benefits and disadvantages of different supraglacial lake depth estimation techniques, while demonstrating that relatively reliable estimations can be obtained through more simplistic methods under certain conditions.

## Appendix A: Data acquisition information

**Table A1: ICESat-2 and Sentinel-2 data used for the development of the ICESat-2-based lake depth algorithm in this study.**

| Date | ICESat-2 Beam ID | Number of Lakes | Sentinel-2 Tile ID |
|---|---|---|---|
| 1 August 2019 | gt3l | 1 | S2A_MSIL2A_20190801T153911_N0208_R011_T27XVH_20190801T185645 |
| 14 August 2019 | gt2l, gt3l | 2 | S2B_MSIL2A_20190813T152819_N0208_R111_T22WED_20190813T185854<br>S2B_MSIL2A_20190814T150019_N0208_R125_T22WEB_20190814T183619 |
| 18 August 2019 | gt1l, gt3l | 2 | S2B_MSIL2A_20190817T150809_N0208_R025_T22WEB_20190817T202054<br>S2B_MSIL2A_20190817T150809_N0208_R025_T22WEA_20190817T202054 |
| 6 July 2020 | gt2l | 3 | S2B_MSIL2A_20200705T151809_N0209_R068_T22WEB_20200705T185648 |
| 29 June 2021 | gt2r | 1 | S2A_MSIL2A_20210628T152911_N0300_R111_T27XVJ_20210628T191004 |
| 2 August 2021 | gt2r, gt3r | 3 | S2A_MSIL2A_20210801T150911_N0301_R025_T22WEB_20210801T171130<br>S2A_MSIL2A_20210801T150911_N0301_R025_T22WEA_20210801T171130 |
| 26 August 2021 | gt2r, gt3r | 2 | S2B_MSIL2A_20210826T150759_N0301_R025_T26XNN_20210826T185448 |
| 31 July 2022 | gt2r, gt3r | 3 | S2B_MSIL2A_20220801T150809_N0400_R025_T22WEB_20220801T185623<br>S2B_MSIL2A_20220801T150809_N0400_R025_T22WEA_20220801T185623 |
| 4 August 2022 | gt1r | 2 | S2B_MSIL2A_20220804T151809_N0400_R068_T26XNN_20220804T190239 |

**Table A2: The acquisition date of the in situ sonar measurements along with the Sentinel-2 image against which the depth data was correlated.**

| Lake ID | Sonar Acquisition Date | Sentinel-2 Tile ID |
|---|---|---|
| 469 | 9 July 2022 | S2B_MSIL2A_20220708T152819_N0400_R111_T27XVH_20220708T174327 |



| 522 | 9 July 2022 | S2B_MSIL2A_20220708T152819_N0400_R111_T27XVH_20220708T174327 |
| 610a | 4 July 2022 | S2A_MSIL2A_20220703T152821_N0400_R111_T27XVH_20220703T202516 |
| 610b | 9 July 2022 | S2B_MSIL2A_20220708T152819_N0400_R111_T27XVH_20220708T174327 |

**Table A3: Data used in the formation of the DEM method comparison analysis. For each lake, the date on which it is first seen drained is listed, along with the date on which the TanDEM-X data was acquired. The Sentinel-2 tile IDs that were used as the basis for the sonar equation, ICESat-2 equation, and the RTM method are also listed.**

| Lake ID | Drainage Date | DEM Date | Sentinel-2 Tile ID |
|---|---|---|---|
| 469 | 1 August 2021 | 13 August 2021 | S2B_MSIL2A_20210730T151809_N0500_R068_T27XVH_20230123T222526 |
| 562 | 24 July 2021 | 13 August 2021 | S2A_MSIL2A_20210721T153911_N0500_R011_T27XVH_20230526T204315 |
| 741a | 21 July 2021 | 23 July 2021 | S2B_MSIL2A_20210720T151809_N0500_R068_T27XVH_20230126T233654 |
| 741c | 20 / 21 July 2021 | 23 July 2021 | S2A_MSIL2A_20210719T145921_N0500_R125_T27XVH_20230126T165617 |
| 741d | 20 / 21 July 2021 | 23 July 2021 | S2A_MSIL2A_20210719T145921_N0500_R125_T27XVH_20230126T165617 |

**Appendix B: Additional data analysis**

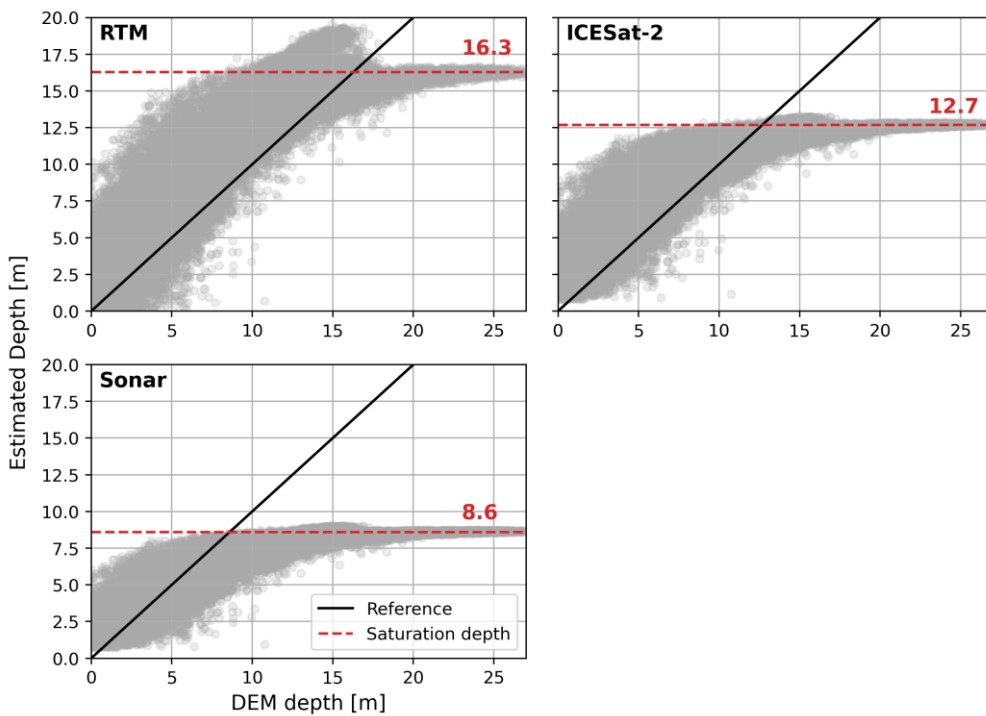



**Figure B1: The depths [m] derived from the DEMs of all five lakes (469, 562, 741a, 741c, and 741d) against the estimated depths [m] from the RTM method, ICESat-2 equation and the sonar equation. A reference line (black) is given to represent where the DEM and estimated depths would be equal. A red dashed line shows the depth at which each method is saturated (stays stagnant) even though the DEM depth increases.**

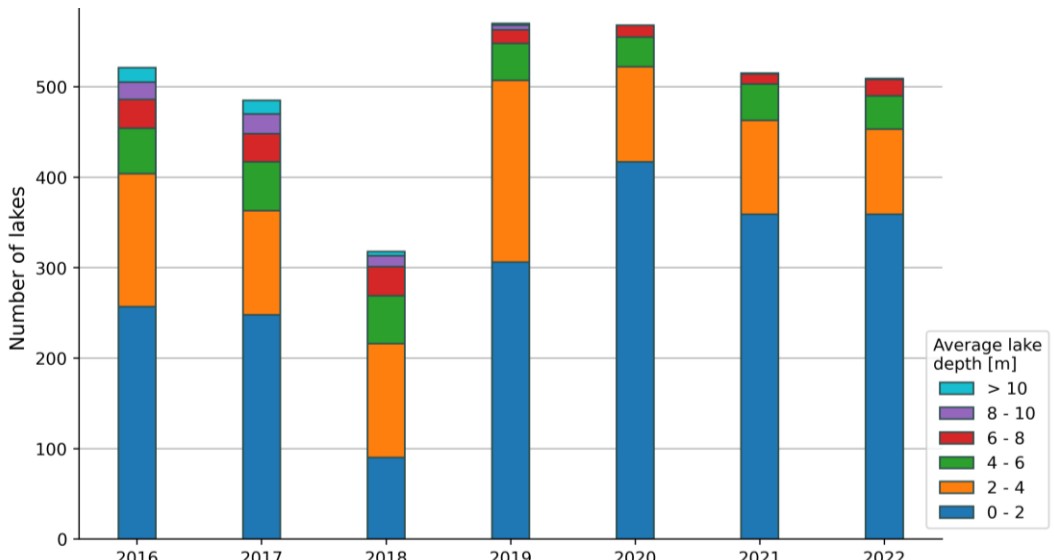

**Figure B2: Number of lakes on the yearly peak melt dates, categorized by the average depth [m] of each lake calculated from the sonar method.**

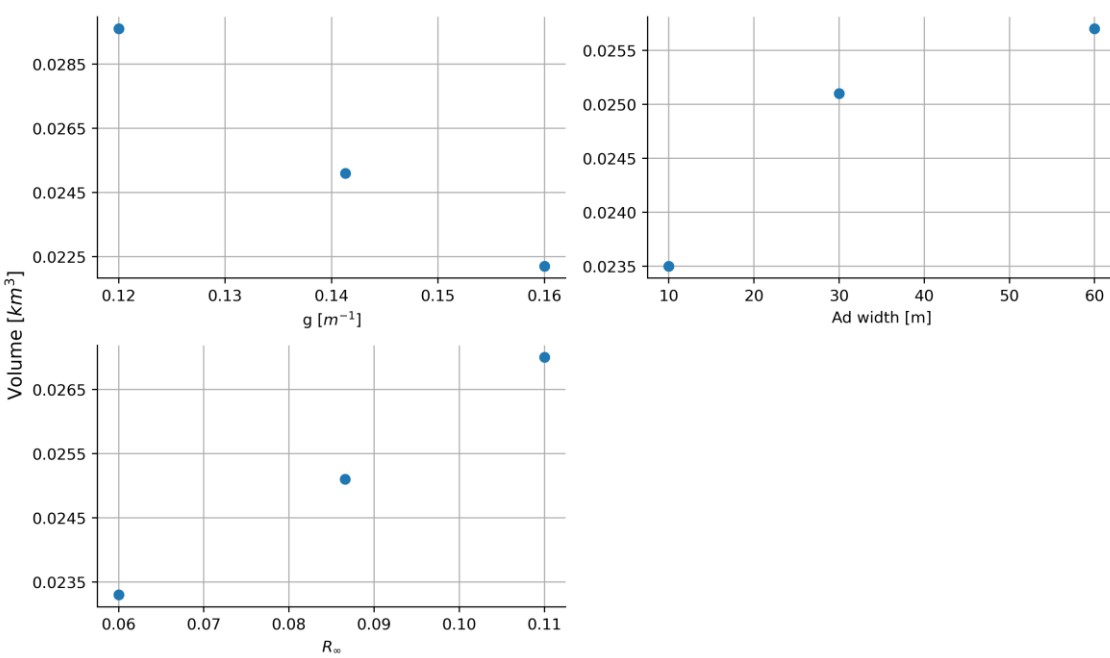

**Figure B3: The sensitivity of the two-way attenuation coefficient, $g$, the width of ice around the lake considered in calculating the lake bed albedo, $A_d$ width [m], and the reflectance of optically deep water, $R_\infty$. The estimated lake volume [km$^3$] for Lake 562 is given for the different variable inputs. The central value for each parameter was used in the method comparison in Sect. 3.3.**



*Code and data availability*: Any code or data produced in this study is available upon request.

*Author contribution:* KL and MB were responsible for conceptualization. KL was responsible for data curation. KL and LB
developed the methodology and conducted the formal analysis. The investigation, specifically the collection of field data, was
conducted by KL, MB, AH, and MS. Resources were procured by KL and MB. KL and CS were responsible for the
development and implantation of necessary software. MB was responsible for funding acquisition, project administration, and
supervision. KL was responsible for data visualization and the preparation of the original draft. All co-authors contributed to
reviewing and editing the manuscript.

*Competing interests:* The authors declare that they have no conflict of interest.

*Acknowledgements:* We would like to acknowledge the support of the Alfred-Wegener-Institute by granting us space aboard
the Polarstern PS131 expedition to conduct research. We would also like to thank ESA, DLR, and NASA for providing the
Sentinel-2, TanDEM-X, and ICESat-2 data, respectively, free of charge. We also acknowledge financial support by the
Deutsche Forschungsgemeinschaft and the Friedrich-Alexander-Universtät Erlangen-Nürnberg within the funding program
"Open Access Publication Funding."

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
