# Peer review of "Assessing supraglacial lake depth using ICESat-2, Sentinel-2, TanDEM-X, and in situ sonar measurements over Northeast and Southwest Greenland"

_EGUsphere, 2024_

## Referee Comment (RC1)

Review of "Assessing supraglacial lake depth using ICESat-2, Sentinel-2, TanDEM-X, and in situ sonar measurements over Northeast Greenland"

General Comments

Thank you for submitting this research - it is a lovely piece of work which eloquently and almost completely defends the research claims of the authors. I believe the overall quality of the manuscript meets the expectations laid out in the principal criteria of TC. In particular, the in situ sonar data will aid other researchers and constitutes an important addition to the knowledge base. Overall, I only have a few suggestions for the authors which I expect will constitute large enough changes to the manuscript to be of use.

Specific Comments

*Title*

1) I suggest the addition of "southwest" in the title considering the foray into SW Greenland as shown in Fig. (1) i.e. "Assessing supraglacial lake depth using ICESat-2, Sentinel-2, TanDEM-X, and in situ sonar measurements over northeast and southwest Greenland"

*General*

2) Although this is a personal preference, the use of the active voice as opposed to the passive voice would greatly aid the readability of this manuscript and make it substantially more engaging.

3) Please refer to the radiative transfer equation as either 'equation', 'model' or 'algorithm', rather than a mix of those options as it will make it easier to search once the manuscript is published (and it is easier to understand what you are referring to).

4) When referring to the components of the radiative transfer equation, please use italics as this is the way they are referred to in the rest of the literature and it is the way TC asks for mathematical notation to be used in-line.

5) Did you produce the TanDEM-X DEMs yourself or were they acquired as DEMs? Your manuscript suggests you produced them, but your acknowledgements suggest you got them as complete products.

*Introduction*

6) [line 45] Philpot (1987) is the correct citation for this equation – see in References section of this review.

7) [line 46-48] I strongly suggest citing Melling et al. (2024) here (and perhaps elsewhere in the manuscript). The Melling et al. (2024) paper is very relevant to this manuscript and compares the use of the same equation by Philpot (1987), ArcticDEM digital elevation modelling, and ICESat-2 laser altimetry. See References for the full citation.

8) [line 57] Please explain what 'optically deep water' constitutes – I think it is 'more than 40 metres deep' from the Sneed and Hamilton papers cited.

9) [line 59] Change "Ice, Cloud and Elevation Satellite" mission to "Ice, Cloud and Elevation Satellite 2 mission"

10) [line 60] Change to "Lake Surface-Bed separation algorithm (Fair et al., 2020) and Watta algorithm (Datta and Wouters, 2021)".

11) [line 62-63] Suggest removal of "go one step further" as this is not required.

12) [line 69] "dramatic perturbations" feels like odd phrasing, perhaps change to "large perturbations"

13) [line 71] Currently, you define the RTM abbreviation on this line but it would be helpful if you define it at the first instance of the term. This occurs on line 43.

14) [line 73-75] Suggest changing to "The associated errors and uncertainties of each method are quantified and discussed to understand the pitfalls of each method." (with the potential change to the active voice as discussed in comment #2).

*Data and Methods*

15) [line 78-79] Suggest changing to "This study consists of four methods based on various data sources." The end of this sentence does not add to the manuscript and can be removed without issue.

16) [line 83] Add a citation for the revisit time.

17) [line 92-93] Suggest changing to "Ideally, suitable imagery is acquired from the same date as the ICESat-2, in situ sonar, or TanDEM-X data; …"

18) [line 98] Unless the rest of the depths are written as negative values, this equation should read as "$g$" under the fraction line, not "- $g$". Suggest changing to "$g$" considering the rest of the manuscript.

19) [line 104] Cite Moussavi et al. (2020) here for 30-m ring for $A_d$ (already in your reference list).

20) [line 105-106] Suggest changing to "Furthermore, although it is intended that $R_\infty$ be calculated for each image, optically deep water is not present in every scene due to …"

21) [line 106-110] Melling et al. (2024) lists the relative importance of the RTE parameters which should help defend your calculation of $R_\infty$ here.

22) [line 110] I strongly suggest adding a sentence or two here referring to Melling et al. (2024) and their comparison of the multiplier on the $K_d$ value as $g$ holds a lot of sway in the outcome of this equation.

23) [line 117] Add Das et al. (2008) citation for rapid SGL drainage (already in your reference list).

24) [line 118] Suggest changing "enabled" to "possible" as the current sentence reads strangely.

25) [Figure 1] I would be personally interested to know if any of the studied lakes in the southwest are the same as the lakes studied in Melling et al. (2024) (no need for a change here, just scientific curiosity for future studies)

26) [Figure 1] I suggest having the ICESat-2 beam IDs in the Appendix only, perhaps link to the ID number with e.g. "NE1", "NE2", "SW1" etc. This should make the figure easier to read.

27) [line 146] Please add some citations here to back up "Based on best judgement and consistency with previous studies".

28) [line 161-162] $n_1$ and $n_2$ values are taken from Mobley (1995) – please reference the original paper as opposed to implying this is Parrish et al. (2019). Equally, the value for $n_2$ is 1.33469, not 1.334. Please redo the analysis of this lake depth equation with 1.33469 before resubmitting. See reference list for citation.

29) [line 171] From the Climate Change Initiative (see References) none of these lakes are upstream of the 2017 grounding line for Zachariae Isstrom. As such, the area will rise and fall with the tides and cannot be considered grounded. However, this shouldn't pose too much of a problem as you seem to have used depth relative to the surface as opposed to absolute depth. Saying this, assuming these lakes are grounded has given you the wrong narrative for these lakes – their dynamics will be different and this will have affected your interpretation of the results later on in the manuscript. Please take some time to consider the effect of this understanding change and alter the manuscript accordingly. In my expert knowledge on the topics of lake depth and grounding lines, this should not invalidate your in situ data.

30) [line 175] How did you estimate your error of 0.20 cm? I would like to see a sentence or two added here to explain your calculation.

31) [line 185] Change to "… captured one day before…" instead of "…captured from the date before…" – this is a clearer way of explaining your data acquisition dates.

32) [Figure 4] It would be good to see a discrete colour bar here instead of a continuous one. I suggest colour steps of one metre. It will not drastically reduce the depth resolution of the plot but would make the figure substantially easier to interpret as the reader.

33) [line 211] Change "is a relatively rare event" to "is difficult" or an appropriate synonym of "difficult".

34) [line 218] Remove "Nonetheless".

35) [line 221] Change "In order to" to "To".

*Results*

36) [line 236] Change "reference" to "represent" for both instances on this line. "Reference" feels misleading.

37) [line 242] I think you are referring to Fig. 5(b) here, not 5(c).

38) [line 243] Same conclusion was reached by Melling et al. (2024). Adding this should add weight to your claims.

39) [line 251, Figure 5 caption] $R^2$ looks strange here, it is not the same as in other parts of the manuscript.

40) [line 252] See comment above.

41) [Figure 6] I suggest using a discrete colour bar here instead of continuous for the same reasoning as the comment on Figure 4. Please also add a north arrow to each of the top row panels.

42) [line 315] Suggest changing "This large difference can be seen for example" to "An example of this large difference is seen"

43) [line 328, Figure 7 caption] Remove "[km$^3$]" this is not required as it is in your figure.

*Discussion*

44) [line 334] I think you mean 3 metres here, not 4 metres.

45) [line 356] Insert "of" between "couple" and "effects".

46) [line 417-419] Suggest moving sentence starting with "Furthermore" and ending in "overestimated" to the end of line 411. This helps it to read better. If you agree, also change:
   a. [line 412] "second" to "third"
   b. [line 419] "Contrarily" to "Similarly"

47) [line 419] Change "thus" to "also".

48) [line 420] Remove "also".

49) [line 421-422] The sentence starting "Finally" is basically a reproduction of the part starting on line 412. Suggest removing "Finally, the presence of floating ice…" sentence and moving the part starting in "This also can be problematic…" to the sentence ending in "deep part of a lake to a shallower part." To the end of the sentence which finishes on line 417.

*Appendix A*

50) [Table A1, column 2] Please make the ICESat-2 Beam ID into the full ICESat-2 file path for reproducibility.

51) [Table A1] I suggest the addition of another column that lists the location of the lake (by region i.e. NE, SW, CW)

52) [Figure B1, caption] Remove reference to "[m]" (2 instances) in caption as these are in your figure.

53) [Figure B2, caption] Remove reference to "[m]" (1 instance) in caption as this is in your figure.

54) [Figure B3] Include subplot identifiers e.g. a, b, and c or $g$, $A_d$, and $R_\infty$

55) [Figure B3, caption] Remove reference to "[km$^3$]" as this is in your figure.

**References**

Climate Change Initiative (ESA Greenland_Icesheet_CCI, Grounding Lines from SAR Interferometry) link: http://products.esa-icesheets-cci.org/products/details/greenland_grounding_line_locations_v1_3.zip/

Melling, L., Leeson, A., McMillan, M., Maddalena, J., Bowling, J., Glen, E., Sandberg Sørensen, L., Winstrup, M. and Lørup Arildsen, R., 2024. Evaluation of satellite methods for estimating supraglacial lake depth in southwest Greenland. The Cryosphere, 18(2), pp.543-558.

Mobley, C.D., 1995. The optical properties of water. Handbook of optics, 1(43), p.43.

*Further comments*

I wholeheartedly believe that reference to the Melling et al. (2024) paper will substantially reinforce the findings detailed in this manuscript. However, I understand it may come across as citation manipulation so I suggest that the authors only cite this paper if they agree that reference to it will reinforce this manuscript. I can be contacted at l.melling@lancaster.ac.uk and would be willing to expand on any of the above comments if required.

---

## Referee Comment (RC3)

**Assessing supraglacial lake depth using ICESat-2, Sentinel-2, TanDEM-X, and in situ sonar measurements over Northeast Greenland**
Lutz et al.

In this manuscript, the authors compare 4 different methods for calculating lake depth. The first two methods empirically fit optical reflectance values to depth measurements obtained from ICESat-2 and in-situ sonar measurements. The third method (RTM) relies solely on optical observations, using a physically based radiative transfer equation to estimate lake depth. The final method uses DEM differencing. The authors demonstrate and discuss the limitations of using each method to estimate supraglacial lake depths. Overall, this work is useful and will be a valuable addition to the scientific community. I have no major concerns with the methodology. While my list of comments my appear long, I believe most of them are relatively minor and I applaud the authors with their efforts on this work.

**General comments**

I think that the writing can be improved throughout. Generally, there are places where many words can be replaced with a few words to improve readability and flow. For example:

- L20: remove "able to be" from "were able to be procured"
- L48: "research has been conducted to fit empirical functions…" can be replaced with "empirical functions have been fit…"
- L51: remove "in their analysis"
- L60: "begun to be" can be replaced with "been".
- L84: Replace "the monitoring of such a dynamic hydrologic process" with "monitoring dynamic hydrologic processes".
- L88: Replace "the presence of atmosphere in an image" with "the atmosphere".
- L105: Replace: "due to the fact that" with "as"
- L108: "the values of g are" can be changed to "g is"
- L121: Replace "in order to allow" with "To obtain"
- L139: "lake crossings of SGLs" is redundant.
- L261: Replace "across the reflectance values" to "with reflectance".

These are just a few examples where the writing is redundant. Please check the rest of the manuscript for sentences where the writing can be similarly condensed.

I also find some phrases and sentences to be vague. For example:

- L65 – Which "several other algorithms"?
- L69 – "within a certain margin" – what is this margin? Can this be quantified?
- L150 – "preprocessed by converting the digital numbers to reflectance values" – what is meant by this?
- L200 – "configurable high resolution spaceborne radar…" What is meant by 'configurable'? What is the resolution?

- L244 – "data points for the green band only start stacking…" What is meant by 'stacking' here?
- L382 – "The ocean itself also inherently has a relatively wide spread of reflectance values" How wide?

Keep an eye on consistency. Sometimes 'SGL' is used, and other times 'supraglacial lake' is used. Sometimes radiative transfer model is used and other times it is 'algorithm'. Sometimes 'RGB' is used and other times this is spelt out. There are also places in the methods where the tense switches between present and past (L120-125, L170-172, L215-220, L340-343). I would recommend writing the methods section in past tense, although this may just be a personal preference. Finally, most of the manuscript is written in a passive voice however a few sentences use first person pronouns (L121, L205, L334, L339). Personally, I think writing in an active, first-person voice is clearer and more concise. I would encourage the authors to use active voice more frequently, especially throughout the methods section, or to remove the few instances with first-person pronouns.

Finally, be careful of sentences that begin with "this", without specifically stating what "this" refers to. Some examples include L194, L238, L263, L319, L422.

**Abstract**

The tense changes from past to present and back again several times in the abstract.

L14 – change 'regression' to 'empirical'

L14-18: "The first empirical…. to create empirical relations". I think that these sentences should be simplified to something like: "The empirical methods are developed to relate Sentinel-2 reflectance values to supraglacial lake depth obtained from 1) ICESat-2 crossings over 19 lakes in Northeast and Southwest Greenland, and 2) in-situ sonar tracks from four lakes on Zachariæ Isstrøm in Northeast Greenland."

L20: remove "able to be"

L21: Add "empirical" between "sonar-based" and "equation"

L22: "Through the evaluation… lake bed sediment could be seen". This sentence is a bit vague and feels out of place in the abstract. Please either expand a bit (e.g. what is the influence of the lake bed sediment?), or remove.

L23: Replace "appropriately adapted equation" with "ICESat-2 empirically derived depth equation"

L27: Add "to explore lake volume interannual variability" or something to the end of this sentence to elaborate on why this was done from 2016-2022.

L28: Consider replacing "pitfalls" with "limitations"

The last sentence of the abstract is a bit weak, specifically "while retaining sufficient accuracy under certain conditions". What does "sufficient accuracy" mean? Under what conditions? I have the same issue with the final sentence of the manuscript.

**Introduction**

L34 – replace "variations" with "variability".

L35 – replace "developmental rate of the lakes" with "timing of lake development".

L35 – I think that "topographical depression" is a more commonly used phrase than "surface sink". For the term "sink" I usually think of something that takes up some resource (e.g. carbon sink). Consider replacing this phrase elsewhere as well (L176).

L36 – I think the comma after "locations" can be removed.

L37 – Remove "allowing for lake development to be easily tracked."

L38 – Here, it may be better to specifically cite Greenland-based work. Now mostly Antarctica-based work is cited (Arthur et al 2020, Dirscherl et al 2020, Dell et al 2021, Corr et al 2022). There is a lot more Greenland-based work that could be cited here (e.g. Williamson et al 2017, Miles et al 2018, Hu et al 2022, Dunmire et al 2022, Zhang et al 2023, …).

Paragraph 1 – In this paragraph there is a missing the link between the surface and subglacial hydrologic networks. Ie: how does water get from the surface to the bed? I would recommend adding a sentence or two on hydrofracture, moulins, etc, and how the water gets from supraglacial lakes to the subglacial hydrologic system.

L43 – Maybe change "presented" to "explored" or "commonly utilized"?

L43 – Change "This method" to "The radiative transfer model"

L45 – Add "water" before "depth".

L46 – Replace "implemented on SGLs by many research groups on various data sources and areas of interest" with "commonly used to estimate SGL volume across the GrIS". And again, I would focus your citations on Greenland-based work (could add Glen et al 2024, MacDonald et al 2018).

L55 – I would recommend combining the last two sentences of this paragraph. For example: "Furthermore, these methods are limited by ….". Also does "these methods" specifically refer to the empirical methods or to all methods which utilize optical imagery? Please specify for clarification.

L67 – Replace "physically based algorithm" with "radiative transfer model" for consistency and please check other places as well.

Somewhere in the introduction I think sonar data should be introduced (e.g. what it is and how it can obtain lake depths).

L75 – How are the errors and uncertainties quantified? What "truth" are the methods compared against? After reading the full manuscript I don't think the errors of the methods are truly quantified so I would consider rewording this sentence.

The introduction is missing a citation and discussion of Melling et al 2024. How does this work complement and expand upon what was done in that work?

**Data and Methods**

In general, I think the organization of this section should be re-worked. The subsections switch between data, methods, and study region. Maybe it would make sense to break up the Data and Methods into separate sections? I think a more organized approach would start by introducing all the data used and then move on to the 4 different methods.

How are the lakes in this study delineated? Is this done manually, or have you used a pre-existing algorithm?

L78-80: I would consider removing these lines. They feel out of place considering they introduce the four methods and then the next subsection immediately covers the data.

L85: Change "rendering" to "which renders"

L86: "which is valuable for a detailed analysis of the lakes". This depends a bit on the context… the 10m resolution of S2 is certainly an improvement compared to Landsat but is still not fine enough to resolve some features such as smaller fractures which can be resolved in WorldView imagery, for example.

L91: Replace "cloudy days" with "cloudy images".

L91: "solely or in combination with other data sources" is unnecessary.

L93: "other data acquisitions" is a bit vague since the other data sources have not been described yet. I would add in parenthesis (e.g. ICESat-2 and in-situ sonar data)

L95: Please specify the maximum date offset for S2 imagery, compared with the other data sources.

L106: remove "here" before R_inf

L107: remove "empirically" and add "the reflectance of" before "optically deep water".

L110: Other work has combined the red and green bands for lake-depth estimation. Can you elaborate why only the green band is used here? I know this comes a bit later on in the discussion, but it may also be helpful reasoning here in the methods section.

L115: Consider adding a brief description of the ATL03 product.

L116: Replace "close" with "intraseasonal"

L117: The wording of the last two sentences in this paragraph is a bit awkward, especially "such a monitoring task is enabled".

L120: Surely these lakes are not the only lakes where an ICESat-2 path crossed a filled SGL? Did you look at all GrIS regions or just NE and SW?

L149: Replace "significant" with "substantial", as I assume there was no actually significance test for this?

L149: The sentences beginning with "Figure 2(b)…" seem out of place in this ICESat-2 lake cross tracking retrieval section, as these sentences refer only to Sentinel-2 data.

L163 – I would consider modifying this sentence: "Then, the corrected depths and…" to something like: "The corrected ICESat-2 depths are then compared with RGB reflectance values for all 19 lakes. An exponential function is fitted to each band and the $R^2$ values are used to determine which optical band best correlates with lake depth.", or something of the sort.

L167: Replace "One of the depth algorithms" with "The second empirically-derived depth algorithm"

L175 – How is this error estimated? Is this from a different paper?

L175 – The naming convention is unclear to me. How is it based on the location of the the topographical depressions? As there are only 19 lakes in the manuscript here,

perhaps it would be clearer to rename the lakes to something simpler (thus also removing the need for both numbers and letters in the naming).

L186 – Please include which lakes these tracks are for.

L191 – Perhaps it is helpful to say here that this process (the creation of the depth-reflectance relationship) was done similar to how it was with ICESat-2. In L191, I think it is important to mention that this is done for each S2 band.

L208 – Why are the DEMs used from after lake drainage? Would it make more sense to use DEMs from before lake-filling as this would more accurately represent the non-lake surface? After drainage, the DEM surface may also include ice fractures or ridges that result from the drainage, which would therefore provide an inaccurate representation of the actual lake depth.

L218 – How is the surface elevation RMSE calculated? Is it compared with the Copernicus DEM? Or is this really the standard deviation of the lake edge pixel elevation?

**Results**

L235 – Replace "reference" with "represent".

L236 – I think it should be specifically said that the ice in SW Greenland has a "lower albedo".

L257 – Remove "gathered"

L258 – Specify that this is RMSE of the exponential fit.

L260-265 – I don't fully follow this section. For example: "while the data points in the Southwest function…". Are these 'data points' from actual ICESat-2 data? The use of 'in the Southwest function' confuses me a bit.

L271 – Out of curiosity, if Lake 469 was included in the analysis, would it still reasonably fit the same curve in Fig. 5d?

L280 – It is mentioned that the RMSE ranges from 0.27 to 0.94 m, but for which reflectance values is the RMSE relatively low or high?

Section 3.3 – A figure that shows depth error (compared with DEM) with lake depth for each non-DEM method would be helpful to reference throughout this section. For example, I'm thinking of something like this below:

[Figure]

L286 – What are the errors associate with using the DEM results as a reference? How accurate are the DEM results?

L288 – In comparison, what is the maximum depth determined from the DEM method?

L288 – The sentence: "The sonar equation produces the largest errors in the shallower regions", reads to me as: compared to the other methods, sonar is the worst in shallow water. Instead, I guess it is meant that sonar does worse in shallow water compared to its performance in deeper water. Consider rewording this for clarity.

L303 – What does 'significantly' mean here? 1m? 5m?

L312, Figure 7 – How are these uncertainties determined?

L314 – Could also cite Dunmire et al 2021 here.

L322 – add "to" after "comparing".

**Discussion**

Again, a reference to Melling et al 2024 and a discussion of the results in the context of this previous work is missing here.

L339 – Specify that the 'this' in "this limits the scope of such a method" refers to the saturation of the red band at higher lake depths.

L341 – "depths between 10 and 25 m." Are these maximum depths?

L341 – "Moreover, in the interannual comparison…" Which method are these statistics based on?

L343 – replace "in the lakes where average depths are larger" with "in lakes with deeper average depths".

L345 – "However, an analysis…" This sentence reads awkwardly and should be reworded for clarity.

L352 – What we care about at the end of the day is the actual volume of water stored in SGLs. The red band underestimates deep depths and the green band overestimates shallow depths; but, we can't truly know which method is better suited without a similar comparison between red-band derived depths and DEM-derived depths. I'm not fully convinced that the question of which band is better suited can be answered with this work as it only includes 5 lakes in NE Greenland. How do we know that these 5 lakes are generally representative of GrIS SGLs?

L358 – Can you quantify how many (or what %) of points used in the regression are deep (> 7m) or shallow (< 0.5m).

L359 – It is not immediately clear to me why not having many very deep or shallow points for the regression would lead the ICESat-2 equation to overestimate depths.

L362 – Change "where the edges of the lake are never estimated to be as shallow as they are in the DEM estimates." to "where lake depth at the lake edge is overestimated compared to the DEM method."

L364 – "… the sonar equation regression never reaching a value below 0.5 m…" Can you place a boundary condition on the regression equation such that the line has to reach 0m in this range of reflectance values? From Figure 5 it appears that there are several points that are near 0m depth. Can you force your regression to cross the y-axis somewhere between these reflection values so that the equation is physically bound?

L387 – "… imperfect lake masks…" Did you consider remasking the lakes to be sure that no water pixels were included in your calculation of Ad?

L390 – Change "could improperly reflect the situation in reality" to "may not be realistic".

L393 – This is a long paragraph and I would recommend starting a new paragraph at the sentence that begins with "Thus…"

L394 – Since you discuss % changes in volume of the lake, I think it would be helpful to provide the parameter changes as % changes as well. E.g: what % change is a 0.01 $m^{-1}$ change in $g$?

I would like to see a slightly expanded sensitivity analysis. Fig. B3 only has 3 points per panel. I think this analysis would be improved if there were more values tested. Also, in L394, it is mentioned that a 0.01 $m^{-1}$ change in $g$ results in a 7.4% change in the lake volume, but none of the points in Figure B3 represent a 0.01 $m^{-1}$ change in $g$.

L400 – "… ice surface is more dynamic, e.g. with sediment dispersion." With the word 'dynamic' here, I think of the ice surface as physically moving. I think changing to 'variable' would be helpful.

L405 – Remove the words "there are" and "which" from this line.

L409 – I prefer the terms "overestimated" and "underestimated" than "deeper/shallower than in actuality" (see L417 as well)

L410 – "Due to the insights gained from the ICESat-2 analysis…" I recommend specifying these insights here.

L415 – Replace "noticeable" with "obvious".

L444 – Please comment on the applicability of the regression methods to a larger area or for continent/basin wide lake volume estimation.

**Figures**
Figure 1 – It would be helpful to also plot the lake locations in 1b. Also, in the caption header the future tense "will be" is used. Please change this to "was". Finally, the titles for the different lakes are not intuitive. I assume the date is in the form dd-mm-yy. I think it would be better to label the images with the date of the optical image (in a more intuitive format), as the beam information is already located in Table A1.

Figure 2 – Include the Sentinel-2 image date explicitly either on the image or in the caption (assuming 18 August 2019 is for the ICESat-2 data?). Remove "plotted" in L154 and "the" before "Sentinel-2 image" in L155.

Figure 3 – Is it necessary to note who took the photos? In panel c, it would be helpful to see this area in relation to the study area in 1b. Also, the borders in panel c seem blurry. In d, I would replace "in situ" with "sonar" in the legend and caption.

Figure 4 – Please include the image acquisition dates either on the image or in the caption.

Figure 5 – I think it would make more sense to order the plots by band wavelength : R (a), B (b), G (c). The legend for the NE and SW Greenland points should be in panel (a). Did you use the Red and Blue bands with Sonar? If so, it may be interesting to see

these bands as panels as well. In the caption, please specify *SGL* depth (L247). How does the fitted curve in d compare with the fitted curve in c?

Figure 6 – Should mention the image date for each lake, either in the image or the caption. A separate (maybe in Appendix) figure showing the depth difference between each method and the DEM method would be helpful to more clearly visualize the differences, specifically as this is discussed in section 3.3.

Figure 8 – Include a map of where these lakes are located.

Figure B1 – Include panel labels (a), (b), (c). I think it would also be helpful to plot these scatter plots on a square grid (with the same limits for x and y-axis).

Figure B2 – The colors for > 10 and 0-2 appear very similar to me.

Figure B3 - Include panel labels (a), (b), (c). It may be better to represent the axis as % changes. So, % change in the parameter values and corresponding % change in calculated lake volume.

**References used in this review**

Dunmire, D., Banwell, A. F., Lenaerts, J., & Datta, R. T. (2021). Contrasting regional variability of buried meltwater extent over two years across the Greenland Ice Sheet. The Cryosphere Discussions. doi: 10.5194/tc-2021-3

Glen, E., Leeson, A. A., Banwell, A. F., Maddalena, J., Corr, D., Noël, B., & McMillan, M. (2024). A comparison of supraglacial meltwater features throughout contrasting melt seasons: Southwest Greenland. The Cryosphere Discussions.

Hu, J., Huang, H., Chi, Z., Cheng, X., Wei, Z., Chen, P., Xu, X., Qi, S., Xu, Y., and Zheng, Y.: Distribution and Evolution of Supraglacial Lakes in Greenland during the 2016–2018 Melt Seasons, Remote Sens.-Basel, 14, 55, https://doi.org/10.3390/rs14010055, 2022.

Macdonald, G. J., Banwell, A. F., & MacAyeal, D. R. (2018). Seasonal evolution of supraglacial lakes on a floating ice tongue, Petermann Glacier, Greenland. Annals of Glaciology, 59 (76pt1), 56–65. doi: 10.1017/aog.2018.9

Miles, K. E., Willis, I. C., Benedek, C. L., Williamson, A. G., & Tedesco, M. (2017). Toward monitoring surface and subsurface lakes on the Greenland ice sheet using sentinel-1 SAR and landsat-8 OLI imagery. Frontiers in Earth Science, 5 (July), 1–17. doi: 10.3389/feart.2017.00058

Williamson, A. G., Arnold, N. S., Banwell, A. F., & Willis, I. C. (2017). A Fully Automated Supraglacial lake area and volume Tracking ("FAST") algorithm: Development and application using MODIS imagery of West Greenland. Remote Sensing of Environment. doi: 10.1016/j.rse.2017.04.032

Zhang, W., Yang, K., Smith, L. C., Wang, Y., van As, D., Noël, B., . . . Liu, J. (2023, 11). Pan-Greenland mapping of supraglacial rivers, lakes, and water-filled crevasses in a cool summer (2018) and a warm summer (2019). Remote Sensing of Environment, 297, 113781. doi: 10.1016/j.rse.2023.113781

---

## Author Comment (AC1)

**Response to Reviewer #1 – Laura Melling, submitted 17 June 2024**

General response

Thank you for the detailed and constructive feedback on our manuscript. We are pleased that our research was positively received and the importance of our findings highlighted. We find the comments made in your review to be beneficial suggestions to the manuscript and have integrated the majority of them. Here, I will address the comments more specifically.

Specific responses

*Title*

1) I suggest the addition of "southwest" in the title considering the foray into SW Greenland as shown in Fig. (1) i.e. "Assessing supraglacial lake depth using ICESat-2, Sentinel-2, TanDEM-X, and in situ sonar measurements over northeast and southwest Greenland"
We agree – we only did not add it originally because the analysis using the derived methodologies was conducted exclusively in northeast Greenland; however, we see that it would be more holistic to include "southwest" as well.

*General*

2) Although this is a personal preference, the use of the active voice as opposed to the passive voice would greatly aid the readability of this manuscript and make it substantially more engaging.
We have revisited the text and integrated the active voice as much as possible for better readability.

3) Please refer to the radiative transfer equation as either 'equation', 'model' or 'algorithm', rather than a mix of those options as it will make it easier to search once the manuscript is published (and it is easier to understand what you are referring to).
We have adjusted the text accordingly.

4) When referring to the components of the radiative transfer equation, please use italics as this is the way they are referred to in the rest of the literature and it is the way TC asks for mathematical notation to be used in-line.
It seems some of the mathematical notation formatting was corrupted during finalization of the manuscript. We have adjusted that accordingly.

5) Did you produce the TanDEM-X DEMs yourself or were they acquired as DEMs? Your manuscript suggests you produced them, but your acknowledgements suggest you got them as complete products.
We indeed processed the TanDEM-X DEMs ourselves. In the acknowledgements, we are simply thanking them for providing the raw TanDEM-X radar data to us.

*Introduction*

We have integrated the adjustments to the text outlined in comments (6) – (14), and we fully agree with including the citation of Melling et al. (2024), and see how the results align with and support our analysis. This citation was simply overlooked since it was published close to the finalization/internal review of our own manuscript.

*Data and Methods*

We have integrated the adjustments to the text outlined in comments (15), (17) – (21), (24), (26) – (27), (31), and (33) – (35).

16) [line 83] Add a citation for the revisit time.

The official stated revisit time for the Sentinel-2 constellation is 5 days at the equator. The statement of "a near daily revisit time in northern Greenland" is based on my own observations and observations of other researchers from using the data and understanding generally how the orbital dynamics of satellites work, i.e. for a polar-orbiting satellite, the more northern areas have a higher density of tracks and thus a higher revisit time. Due to this, there is no specific citation I would add for this statement.

22) [line 110] I strongly suggest adding a sentence or two here referring to Melling et al. (2024) and their comparison of the multiplier on the $K_d$ value as $g$ holds a lot of sway in the outcome of this equation.
We did not add the Melling et al. (2024) citation here, but have gladly added a sentence describing the effect seen from your alteration of $K_d$ and $g$ values into section 4.2 in the discussion.

23) [line 117] Add Das et al. (2008) citation for rapid SGL drainage (already in your reference list).
We find that, especially for such a broad statement, specifically citing one paper seems unnecessary and would be cherry-picking since many papers cover the topic of rapid drainages.

25) [Figure 1] I would be personally interested to know if any of the studied lakes in the southwest are the same as the lakes studied in Melling et al. (2024) (no need for a change here, just scientific curiosity for future studies)
From what I can tell without the exact coordinates, I do not think any of the lakes are the same as in Melling et al. (2024). We do have one ICESat-2 track in common (ATL03_20200706005932_01630805_003_01_gt2l), but I believe the lakes we used are located a bit more north from yours. It would have definitely been interesting to see the direct comparison of our studies.

28) [line 161-162] $n_1$ and $n_2$ values are taken from Mobley (1995) – please reference the original paper as opposed to implying this is Parrish et al. (2019). Equally, the value for $n_2$ is 1.33469, not 1.334. Please redo the analysis of this lake depth equation with 1.33469 before resubmitting. See reference list for citation.
In this study, we did not use 1.33469 for $n_2$ since that was calculated as the refractive index for light with a wavelength of 540 nm. Since Sentinel-2's green band has a nominal wavelength of 560 nm, I had recalculated $n_2$ for that wavelength, which ends up being 1.3343. I indeed did not write the full value in the paper and will correct that, along with adding the Mobley (1995) citation. In the end, the difference in the corrected depths when using 1.3343 or 1.33469 is on the millimeter scale, which is well under the uncertainties from the data and methodology itself.

29) [line 171] From the Climate Change Initiative (see References) none of these lakes are upstream of the 2017 grounding line for Zachariae Isstrom. As such, the area will rise and fall with the tides and cannot be considered grounded. However, this shouldn't pose too much of a problem as you seem to have used depth relative to the surface as opposed to absolute depth. Saying this, assuming these lakes are grounded has given you the wrong narrative for these lakes – their dynamics will be different and this will have affected your interpretation of the results later on in the manuscript. Please take some time to consider the effect of this understanding change and alter the manuscript accordingly. In my expert knowledge on the topics of lake depth and grounding lines, this should not invalidate your in situ data.
Thank you for bringing up this point. You are indeed correct that the 2017 grounding line presented in the Climate Change Initiative is well above the lakes that we measured in situ. This information, however, is contradicted in many other studies. After the large break-up of Zachariae Isstrom's floating tongue between 2002 and 2012, the grounding line has been quite close to the calving front itself. With these grounding lines, our in situ data was all gathered in grounded areas. For examples of

literature showing the lower grounding line, please see Mouginot et al. (2015) and An et al. (2020). It is quite concerning why ESA's CCI product is so off from what others report.

30) [line 175] How did you estimate your error of 0.20 cm? I would like to see a sentence or two added here to explain your calculation.
Thank you for this comment. I realized upon re-reading it, that the units on the error were incorrectly written. I have now corrected it to 0.20 m instead of cm. To address your point concerning the estimation of this error, we assume an error of 20 cm based simply on experience using the sonar tool and visual estimation of the clarity of the bed and surface boundaries. We have now added this clarification into the manuscript.

32) [Figure 4] It would be good to see a discrete colour bar here instead of a continuous one. I suggest colour steps of one metre. It will not drastically reduce the depth resolution of the plot but would make the figure substantially easier to interpret as the reader.
The depths are actually already plotted discretely using one-meter steps. I chose to represent the color bar as a continuous one however to have a more compact view of the scale, since the discrete scale over 14 meters is quite long without a lot of distinction between directly neighboring colors.

*Results*

We have integrated the adjustments to the text outlined in comments (36) – (37), (39) – (40) and (42) – (43).

38) [line 243] Same conclusion was reached by Melling et al. (2024). Adding this should add weight to your claims.
We did not reference Melling et al. (2024) here in the results, but have added several references to the conclusions made in this manuscript in the discussion.

41) [Figure 6] I suggest using a discrete colour bar here instead of continuous for the same reasoning as the comment on Figure 4. Please also add a north arrow to each of the top row panels.
Same as in comment 32: The depths are actually already plotted discretely using one-meter steps. I chose to represent the color bar as a continuous one however to have a more compact view of the scale, especially with such large depths.

*Discussion*

We have integrated the adjustments to the text outlined in comments (44) – (49).

*Appendix A*

We have integrated the adjustments to the text outlined in comments (50) – (55).

**References used in this response**

An, L., Rignot, E., Wood, M., Willis, J. K., Mouginot, J. and Khan, S. A.: Ocean melting of the Zachariae Isstrøm and Nioghalvfjerdsfjorden glaciers, northeast Greenland, Proceedings of the National Academy of Sciences, 118, doi: 10.1073/pnas.2015483118, 2021.

Mouginot, J., Rignot, E., Scheuchl, B., Fenty, I., Khazendar, A., Morlighem, M., Buzzi, A. and Paden, J.: Fast retreat of Zachariæ Isstrøm, northeast Greenland, Science, 350, 1357–1361, doi: 10.1126/science.aac7111, 2015.

---

## Author Comment (AC3)

**Response to Reviewer #2 – Devon Dunmire, submitted 20 June 2024**

General response

Thank you for your positive feedback; we are glad that the benefit of our research is well received. Additionally, we appreciate the thorough and constructive comments given in your review. We generally agree with your comments and have gladly integrated the majority of them into our manuscript. More specific responses to your comments can be found below.

Specific responses

***General comments***

We have made efforts to improve the readability, flow and redundancy of the writing where possible, especially in the examples you noted. We have also paid attention to the consistency of abbreviation usage and have tried to adapt a more consistent tense in the methods section. Overall, we have also tried to change as many phrases as possible to a more active, first-person form where sensible.

***Abstract***

We have integrated the comments mentioned over L14 – 28, in addition to making the tense more consistent and the final sentence more concrete.

***Introduction***

We have integrated the comments mentioned over L34 – 75, with the exception of introducing sonar data in the introduction, as we think the explanation given later in the methods section is sufficient for the reader's understanding.

***Data and Methods***

Any comments not specifically addressed below have been integrated into the Data and Methods section.

"In general, I think the organization of this section should be re-worked. The subsections switch between data, methods, and study region. Maybe it would make sense to break up the Data and Methods into separate sections? I think a more organized approach would start by introducing all the data used and then move on to the 4 different methods."

We had also originally thought to organize it in that way, but after writing it, realized that the sonar and ICESat-2 sections flowed much better when the data was described directly with the methodology.

"How are the lakes in this study delineated? Is this done manually, or have you used a pre-existing algorithm?"
They are automatically delineated using a deep learning method developed in Lutz et al. (2023). This was briefly mentioned in L214 for the TanDEM-X method and L226-227 for the method comparison section, but we agree that it was not made clear that this deep learning method was used throughout the entire method comparison. We have added another sentence to Section 2.6 to clarify that.

"L78-80: I would consider removing these lines. They feel out of place considering they introduce the four methods and then the next subsection immediately covers the data."

We understand how it seems a little jarring introducing the four methods and then going into the Sentinel-2 description; however, we think clearly presenting the four methods (and defining the names we will call them throughout the paper) is beneficial to have at the beginning of the chapter.

"L120: Surely these lakes are not the only lakes where an ICESat-2 path crossed a filled SGL? Did you look at all GrIS regions or just NE and SW?"
There are certainly more lake crossings than the ones we found. We searched for the lake crossings manually and while we disregarded some lake crossings due to low quality surface and bed distinction, there are surely some that were simply missed by us. Due to the high prevalence of SGLs in NE and SW Greenland, we limited our search to those regions; however, one would be able to find crossings in other regions around Greenland as well. A few words were added to this section to hopefully make it clearer that these are just the crossings used in this study – not all possible ones.

"L149: The sentences beginning with "Figure 2(b)…" seem out of place in this ICESat-2 lake cross tracking retrieval section, as these sentences refer only to Sentinel-2 data."
It is true that it does not fit perfectly with the section topic, but to create another section just for a couple sentences seems unnecessary as well. We have made it into a new paragraph and added a sentence before it that highlights the relevance of the next sentences in the context of the section.

"L175 – How is this error estimated? Is this from a different paper?"
I realized upon re-reading it, that the units on the error were incorrectly written. I have now corrected it to 0.20 m instead of cm. To address your point concerning the estimation of this error, we assume an error of 20 cm based simply on experience using the sonar tool and visual estimation of the clarity of the bed and surface boundaries. We have now added this clarification into the manuscript.

"L175 – The naming convention is unclear to me. How is it based on the location of the topographical depressions? As there are only 19 lakes in the manuscript here, perhaps it would be clearer to rename the lakes to something simpler (thus also removing the need for both numbers and letters in the naming)."
We have identified 1035 topographical depressions in our study area in Northeast Greenland and have labelled each one with an ID number. Since we only measured lakes with the sonar boat in Northeast Greenland, we used our depression numbering system to identify them. We have now added a sentence to this paragraph clarifying the system. However, when addressing the lakes with ICESat-2 crossings, we have now changed the identification of these lakes in Figure 1 and Table A1 to more generalized ID numbers (instead of the ICESat-2 beam IDs) since we do not have a system for Southwest Greenland.

"L208 – Why are the DEMs used from after lake drainage? Would it make more sense to use DEMs from before lake-filling as this would more accurately represent the non-lake surface? After drainage, the DEM surface may also include ice fractures or ridges that result from the drainage, which would therefore provide an inaccurate representation of the actual lake depth."
While there certainly could be changes in the ice surface after a drainage due to fracture and uplift, there could also be problems using a DEM from early in the season. The lake could have become buried under a layer of ice over winter, which would create a completely different DEM of the ice surface if the DEM is taken before the surface melts. Furthermore, with a longer time interval between the DEM creation and the lake level comparison, the ice could have flowed substantially, allowing for crevasses or other features to be moved into or out of the lake region. Thus, while there are potentially issues with taking it before or after, we chose to use post-drainage DEMs in this study.

"L218 – How is the surface elevation RMSE calculated? Is it compared with the Copernicus DEM? Or is this really the standard deviation of the lake edge pixel elevation?"
You are right – it is indeed the standard deviation of the lake edge pixel elevations. We have updated that in the manuscript.

**Results**
Any comments not specifically addressed below have been integrated into the Results section.

"L260-265 – I don't fully follow this section. For example: "while the data points in the Southwest function…". Are these 'data points' from actual ICESat-2 data? The use of 'in the Southwest function' confuses me a bit."
Yes, the data points are referring to the data in Fig. 5 (a) – (c), where the depths from ICESat-2 and the reflectance values from Sentinel-2 are plotted against each other for the different bands. For the green band, we fit a curve to the data found in the Northeast and to the data found in the Southwest, which we then refer to as the Northeast function and Southwest function. They are defined in Equations (3) and (4).

"L271 – Out of curiosity, if Lake 469 was included in the analysis, would it still reasonably fit the same curve in Fig. 5d?"
The curve including the data from Lake 469 resulted in the depths estimations being noticeably deeper.

"L280 – It is mentioned that the RMSE ranges from 0.27 to 0.94 m, but for which reflectance values is the RMSE relatively low or high?"
It varies quite a lot over the reflectance values, as it depends on how many data points there are within each bin and how large the spread of values within the bin is. So it is not as simple as saying the RMSE is larger for shallower depths, for example.

"Section 3.3 – A figure that shows depth error (compared with DEM) with lake depth for each non-DEM method would be helpful to reference throughout this section. For example, I'm thinking of something like this below:"
Thank you for this suggestion. We have now created a new figure similar to the example you drew, showing the average error of each method across the DEM depths. We have put this figure in the appendix and reference it throughout Section 3.3.

"L286 – What are the errors associate with using the DEM results as a reference? How accurate are the DEM results?"
The vertical offset of the TanDEM-X DEMs after co-registration, which is the remaining mean vertical deviation between each TanDEM-X acquisition and the Copernicus reference DEM, on stable non-glacierized terrain is 0.17m (SD = 2.40m) and 0.07m (SD = 3.58m) for the 2021-07-23 and 2021-08-13 DEMs, respectively. However, we deliberately did not include this estimate in the manuscript as we do not directly compare surface elevations of the TanDEM-X DEMs to other elevation datasets (e.g. Copernicus DEM or ICESat-2 altimetry). Instead, we extract the relative elevation difference between the lake shore and bed exclusively from each TanDEM-X DEM. Therefore, this relative elevation difference is not related to the accuracy of co-registration but to the creation of the differential interferograms and subsequent phase unwrapping. To avoid biases in the SAR elevation difference due to artefacts during the DEM creation, such as phase jumps, we investigated the distribution of the interferometric coherence, which indicated a loss of coherence for some parts of the lake bed. We removed the respective areas at the lake bottom as shown in Figure 6 to avoid a biased elevation difference estimate due to doubtful DEM values. For the remaining lake bed and shore areas the presence of interferometric distortions is unlikely but unfortunately we cannot quantify a meaningful error budget for the outlined relative elevation difference.

"L288 – The sentence: "The sonar equation produces the largest errors in the shallower regions", reads to me as: compared to the other methods, sonar is the worst in shallow water. Instead, I guess it is meant that sonar does worse in shallow water compared to its performance in deeper water. Consider rewording this for clarity."
We realized this statement is actually misleading and have thus removed it from the manuscript.

"L303 – What does 'significantly' mean here? 1m? 5m?"
The difference in estimation ranges up to 10 m in some areas. This can be better seen in Figure B1 in the appendix.

"L312, Figure 7 – How are these uncertainties determined?"
These uncertainties were determined based on different factors depending on the method and the associated instruments. For the sonar method, we estimated a geolocation error based on the variability between neighboring pixels, an error for wave-rocking of the boat based on crossing points of the sonar tracks, the RMSE of the data spread to the fit curve and the delineation of the lake bed in the sonar reading tool. Similarly, for the ICESat-2 method, we estimated a geolocation error, the RMSE of the data spread to the fit curve, and delineation errors of the lake bed and surface. Finally, for the RTM method, we estimated the error based on the sensitivity analysis shown in Fig. B3, where the change in the parameters of g, $A_d$ width, and $R_{inf}$ over a defined depth change were combined into a single uncertainty. The uncertainties were then applied for each pixel of lake water over the time series and summed per day. More detail to this has now been added to the beginning of Section 3.4.

**Discussion**
Any comments not specifically addressed below have been integrated into the Discussion section.

"Again, a reference to Melling et al 2024 and a discussion of the results in the context of this previous work is missing here."
We fully agree with including the citation of Melling et al. (2024), and see how the results align with and support our analysis. This citation was simply overlooked since it was published close to the finalization/internal review of our own manuscript. Reference to this work has now been added throughout the manuscript.

"L341 – "depths between 10 and 25 m." Are these maximum depths?"
Yes, this was referring to the maximum depths. A few words have been added to the sentence to clarify that.

"L341 – "Moreover, in the interannual comparison…" Which method are these statistics based on?"
They were based on the sonar equation. Since that was indeed not stated, it has now been added to that sentence.

"L352 – What we care about at the end of the day is the actual volume of water stored in SGLs. The red band underestimates deep depths and the green band overestimates shallow depths; but, we can't truly know which method is better suited without a similar comparison between red-band derived depths and DEM-derived depths. I'm not fully convinced that the question of which band is better suited can be answered with this work as it only includes 5 lakes in NE Greenland. How do we know that these 5 lakes are generally representative of GrIS SGLs?"
It is true that five lakes is not a large sample size to make definitive statements on. However, looking at Figure B3, we can see that the average depth of a substantial portion of the lakes found in NE Greenland is above 4 m, where 300 – 550 lakes are assessed for each year. As this is just the average depth of each lake, even lakes with average depths lower than that naturally also have depths reaching higher values. Even though these depths are only estimated using the sonar equation and not DEMs, we have seen with the 5 test lakes that these two methods tend to be in agreement under 8 meters or so. This is a strong indicator that lakes tend to regularly reach depths out of the scope of the red band. That is why we stated in L344 that "…the use of the green band seems to be a more suitable choice for estimating deeper lake depths…". Of course, if it is known that a lake only contains depths up to 3 or 4 meters, then the red band would be a suitable choice. That is also why we state in L449 in the conclusion "to improve the methodology overall, combining estimations from red, green, and blue

bands into a single algorithm could potentially overcome the attenuation limitations of each band, allowing for more accurate estimations in shallow water with the red band…"

"L358 – Can you quantify how many (or what %) of points used in the regression are deep (> 7m) or shallow (< 0.5m)."
For the ICESat-2 data from the northeastern lakes, there are no points above 7 m. For the shallow depths, 24/246 of the data points are below 0.5 m, so 9.76%. For the sonar data, there are only four data points above 7 m (1.64%) and eight data points below 0.5m (3.28%). We have added these statistics to the manuscript.

"L359 – It is not immediately clear to me why not having many very deep or shallow points for the regression would lead the ICESat-2 equation to overestimate depths."
We have removed this phrase from the manuscript.

"L364 – "… the sonar equation regression never reaching a value below 0.5 m…" Can you place a boundary condition on the regression equation such that the line has to reach 0m in this range of reflectance values? From Figure 5 it appears that there are several points that are near 0m depth. Can you force your regression to cross the y-axis somewhere between these reflection values so that the equation is physically bound?"
One could force the regression to reach 0 m; however, we chose not to do this here for a couple of reasons. Firstly, there are only a few points near 0 m and they are spread out over half of the reflectance value range, since the reflectance value in shallow water varies quite drastically due to the influence on the surrounding ice and environmental conditions. Because of this, it would be difficult to choose a reflectance value based on the data presented. Similarly, we wanted the methods to be purely based on the observed data to limit external bias. In future research, it could be valuable to force the equation to reach zero when taking other sources of information into account.

"L387 – "… imperfect lake masks…" Did you consider remasking the lakes to be sure that no water pixels were included in your calculation of Ad?"
This comment was meant in a more general sense, especially when applied to dense time series. Of course for the evaluation of our five test lakes, the lake masks were checked to make sure there were no large errors. That being said, it is not always clear exactly where the edge is. In an automated time series analysis like the one shown in Fig. 7, however, manual evaluation of individual lakes is not feasible. In such cases, if the lake masks are not perfect, the calculation of $A_d$ could be affected.

"L394 – Since you discuss % changes in volume of the lake, I think it would be helpful to provide the parameter changes as % changes as well. E.g: what % change is a 0.01 m-1 change in g?"
The change in volume was represented in percentage because a value given in $km^3$ would not be as directly understandable for the reader since they are not familiar with the overall volume of the lake. A percentage change or, as we additionally wrote, a change in average depth in meters helps the readers put it into context more easily. This, however, would not be the case if the change in $g$ were represented as a percentage, since the readers would be more familiar with the actual values being used for this parameter. Thus, we think it best to leave the representations as they are.

"I would like to see a slightly expanded sensitivity analysis. Fig. B3 only has 3 points per panel. I think this analysis would be improved if there were more values tested. Also, in L394, it is mentioned that a 0.01 m-1 change in g results in a 7.4% change in the lake volume, but none of the points in Figure B3 represent a 0.01 m-1 change in g."
The three values for each parameter were chosen specifically as being the realistic range of values from which one could include in their calculations, with the central value being the one used in our study. Thus, expanding the range further beyond these values would not bring any concrete benefit. Additionally, the parameters $g$ and $R_{inf}$ have seemingly linear relationships to depth, so adding more values would not make the trend any more apparent than it already is. Although the parameter $A_d$

width does not have a directly linear relationship, we also do not think that adding more values between 10 and 60 m would substantially increase the amount of information able to be obtained from the graph. In the end, the focus of this study was intended to show an example of how much these parameters can vary on one lake. To address your last point, after finding the amount the volume changed over the span of $g$ values investigated, we scaled it so that the volume change is represented over a simple unit of $g$ instead of the difference of values shown in Fig. B4.

***Figures***

Any comments not specifically addressed below have been integrated into the figures and/or their captions.

"Figure 1 – It would be helpful to also plot the lake locations in 1b. Also, in the caption header the future tense "will be" is used. Please change this to "was". Finally, the titles for the different lakes are not intuitive. I assume the date is in the form dd-mm-yy. I think it would be better to label the images with the date of the optical image (in a more intuitive format), as the beam information is already located in Table A1."

We have now added the lake locations to subset 1b as well as the glacier labels for better orientation. We have also changed the lake labels to be more easily understandable, i.e. NE1, SW1, CW1 and so on to refer to the region in which they are found.

"Figure 5 – I think it would make more sense to order the plots by band wavelength : R (a), B (b), G (c). The legend for the NE and SW Greenland points should be in panel (a). Did you use the Red and Blue bands with Sonar? If so, it may be interesting to see these bands as panels as well. In the caption, please specify SGL depth (L247). How does the fitted curve in d compare with the fitted curve in c?"

We plotted the bands in that order so that the green band for the ICESat-2 data would be directly next to the green band for the sonar data, allowing for a side-by-side comparison. We did not use the red and blue bands from the sonar data, having seen how ideal the green band was for our purposes in the ICESat-2 data.

"Figure 6 – Should mention the image date for each lake, either in the image or the caption. A separate (maybe in Appendix) figure showing the depth difference between each method and the DEM method would be helpful to more clearly visualize the differences, specifically as this is discussed in section 3.3."

We agree and have added the image dates for each lake to the figure. Since we have added the figure highlighting the difference in depth estimation between the DEM and other methods as you suggested in Section 3.3, we think adding another figure visualizing the differences in depth estimation would be superfluous.

"Figure 8 – Include a map of where these lakes are located."

As these images are used simply as visual examples of observed phenomena, we do not think a detailed map of these specific lakes is necessary.

"Figure B3 - Include panel labels (a), (b), (c). It may be better to represent the axis as % changes. So, % change in the parameter values and corresponding % change in calculated lake volume."

We have now included the panel labels of (a), (b), and (c). We think it is better to not have the data represented in percentages because the specific values of the parameters could be of interest to those looking to use the method themselves.